# Cathepsin B-dependent glycolysis contributes to reduced renal uric acid excretion in hyperuricemia

Huagang Lin [1,2,5], Linjing Nie[1,2,5], Deping Wu[1,2,5], Dexian Zhang[1], Rui Peng[1], Sijie Tao[1], Zhibin Ye[1], Sibo Zhu [3] ✉, Maoqing Ye [2,4] ✉ & Jing Xiao [1,2] ✉

Decreased renal uric acid excretion is a major contributor to hyperuricemia (HUA), but its underlying mechanism remains unclear. Here, we identify cathepsin B (CTSB) as a key regulator of urate handling in HUA. Urinary CTSB levels were elevated in HUA patients, and renal CTSB expression was increased in HUA mice. In CTSB[tecKO] mice, the expression of reabsorptive urate transporters URAT1 and GLUT9 was decreased, while the secretory transporter ABCG2 was upregulated, leading to enhanced renal uric acid excretion and reduced serum uric acid (SUA). CTSB deficiency also reduced serum IL-1β, IL-6, and TNF-α levels. In vitro and transcriptomic analyses revealed that CTSB inhibition suppressed glycolysis—marked by reduced HK2 and PKM2 expression—downregulated URAT1 and GLUT9, and upregulated ABCG2. Conversely, CTSB overexpression enhanced glycolysis and reversed these effects. These findings suggest that CTSB promotes urate retention via glycolysis and may serve as a novel target for HUA treatment.

Hyperuricemia (HUA) is characterized by a significant increase in serum uric acid (SUA > 420 μmol/L). HUA is not only closely associated with gout, and is also a potential risk factor for the progression of chronic kidney disease (CKD), end-stage kidney disease (ESKD), cardiovascular disease, diabetes, obesity, and non-alcoholic fatty liver disease[1–5]. Secondary HUA can be relieved when affecting factors such as diuretic use and volume depletion, etc. are discontinued; however, reduced renal uric acid excretion is the main cause and clinical presentation of primary HUA[6], with molecular mechanisms not clarified.

Renal uric acid transport is reliant on the consumption of ATP by sodium potassium ATPase (NKA) to establish a sodium ion concentration gradient as the primary driving force. This process regulates transport on both sides of the cell membrane, includes regulation of the urate transporter 1 (URAT1)[7]. Approximately 14 uric acid transporters, such as URAT1, work collaboratively in a complex regulatory network to facilitate the excretion of uric acid in urine[8]. Therefore, transport of uric acid in the proximal renal tubule requires energy metabolism. In recent years, numerous studies have found that changes in renal tubular energy metabolism play a crucial role in promoting the progression of various kidney diseases, including acute kidney injury (AKI) and chronic kidney disease[9–12]. However, the mechanism responsible for the abnormal renal regulation of

uric acid metabolism during HUA is rarely studied. In a previous study on the urine exosome protein spectrum of HUA and control patients, we found that lysosome dysfunction was among the most important mechanisms involved in renal tubular injury in HUA patients[13]. Lysosomes are one of the most important organelles in the body and participate in a variety of physiological and pathological processes, including autophagy, macromolecular degradation, exocytosis, and cell death[14]. When lysosomes are injured by HUA, various substances can be released, including cathepsins. Release of cathepsin B (CTSB) is one of the main manifestations of lysosomal injury, and our previous study also found a significant increase in CTSB in renal proximal tubular epithelial cells during HUA. Stabilizing lysosomal membranes with hydroxychloroquine significantly reduced renal tubular CTSB release, inflammation, and URAT1 expression, thus increasing renal uric acid excretion[13].

Increased CTSB is seen in patients with various kidney diseases[15]. Several studies have found that the expression of CTSB in renal proximal epithelial cells is significantly upregulated during HUA[16,17], which was also confirmed by our previous study[13]. In renal proximal tubular epithelial cells under uric acid stimulation, siRNA-mediated CTSB knockdown reduced uric acid-induced inflammation[16]. In addition, kidney CTSB was significantly increased in HUA mice, but after isoorientin treatment, CTSB

[1]Department of Nephrology, Huadong Hospital, Fudan University, Shanghai, PR China. [2]Shanghai Key Laboratory of Clinical Geriatric Medicine, Huadong Hospital, Fudan University, Shanghai, PR China. [3]State Key Laboratory of Genetic Engineering, School of Life Sciences, Fudan University, Shanghai, China. [4]Department of Cardiology, Huadong Hospital, Fudan University, Shanghai, PR China. [5]These authors contributed equally: Huagang Lin, Linjing Nie, Deping Wu. ✉e-mail: sibozhu@fudan.edu.cn; yemaoqing@fudan.edu.cn; jingxiao13@fudan.edu.cn

levels and downstream inflammation were both reduced[17]. These studies suggest that CTSB plays an important role in renal inflammation caused by HUA. However, the effect of CTSB on renal inflammation and renal uric acid excretion is still unclear.

Therefore, our investigation delved into the impact of renal proximal epithelial cell CTSB on inflammation and renal uric acid excretion during HUA. Initially, we scrutinized CTSB alterations in HUA patients, HUA mice, and in cellular context. Subsequently, using CTSB knockout mice, we investigated the effects of CTSB deficiency on interleukin-1β (IL-1β), interleukin-6 (IL-6), tumor necrosis factor-α (TNF-α), URAT1, glucose transporter 9 (GLUT9), ATP-binding cassette subfamily G member 2 (ABCG2), renal uric acid excretion, and SUA levels. Employing RNA-sequencing and metabolomics analysis, we delineated potential mechanisms underlying renal inflammation and aberrant renal uric acid excretion in HUA, subsequently validating these findings both in vivo and in vitro. Lastly, we assessed changes in the overexpression of CTSB in renal proximal tubular epithelial cells.

This study documents a potential role of CTSB in inhibiting uric acid excretion in the proximal renal tubules through glycolysis. The results imply that the pharmacological modulation of CTSB may offer an additional therapeutic avenue, particularly for patients struggling with HUA and comorbidities.

## Results

### CTSB accumulation in the urine of HUA patients and the kidneys of HUA mice

To investigate the changes in the kidneys during HUA, HUA mice were constructed. Compared with the control, the HUA group had higher SUA levels (Supplementary Fig. S1a) and lower fractional excretion of uric acid (FEua) (Supplementary Fig. S1b) but no significant changes in serum creatinine (Scr, Supplementary Fig. S1c) or urinary microalbumin (U-MALB, Supplementary Fig. S1d). Also, serum IL-1β, IL-6 and TNF-α in the HUA group was significantly higher than in the control group (Supplementary Fig. S1e–g). No significant kidney injury was observed by HE staining of the kidney in either group (Supplementary Fig. S1h).

To assess changes in CTSB during HUA, 24-hour urine of control and HUA patients were collected, and CTSB was measured. Patient information is shown in Supplementary Table S2. SUA in the HUA group was significantly higher than in the control group (Fig. 1a). Pearson correlation analysis was performed on SUA levels and FEua in these samples, and a negative correlation was found between SUA and FEua (Fig. 1b). To explore the change in CTSB in patients with HUA, 24-hour urinary CTSB were measured. Interestingly, we found that there was higher urinary CTSB in the HUA group (Fig. 1c), and there was a positive correlation between urinary CTSB and SUA (Fig. 1d).

In the kidneys of HUA mice, both the protein (Fig. 1e, f) and mRNA (Fig. 1g) expression of CTSB increased. To verify the results, another HUA mice (UOX$^{-/-}$) was constructed (Supplementary Fig. S2a). Compared with the SUA of UOX$^{+/+}$ mice, the average SUA of UOX$^{-/-}$ mice was significantly increased (Supplementary Fig. S2b). In addition, Scr and BUN in UOX$^{-/-}$ mice were also significantly increased compared with UOX$^{+/+}$ mice (Supplementary Fig. S2c, d). Consistent with the results from the PO and uric acid drinking water-induced HUA mice, UOX$^{-/-}$ mice also had significant renal CTSB accumulation at both the protein (Fig. 1h, i) and mRNA (Fig. 1j) levels. Finally, these results were further confirmed by immunofluorescence, which showed that the level of CTSB was significantly increased in the renal proximal tubules of HUA mice (Fig. 1k, l).

### CTSB was significantly increased in renal proximal tubular epithelial cells after uric acid stimulation

To verify the results from mice, a human renal proximal tubular epithelial cell line (HK-2) was used. Consistent with the results in vivo, we found that after using 10 mg/dL uric acid to treat cells for 48 h, the expression of CTSB

was significantly upregulated (Fig. 1m, o). Similarly, the qRT-PCR (Fig. 1p) and immunofluorescence (Fig. 1n) results also indicated upregulation of CTSB after uric acid stimulation.

We then repeated the experiment in mouse renal proximal epithelial cells (MRTECs). Not surprisingly, we also observed in MRTECs, the expression of CTSB was also significantly upregulated at the protein level after stimulation with uric acid (Fig. 1q, s). As well, the results of qRT-PCR (Fig. 1t) and immunofluorescence (Fig. 1r) analyses were also consistent with our previous results in mice and in HK-2 cells. These results show that uric acid can induce the accumulation of CTSB in renal proximal tubular epithelial cells in vivo and in vitro.

### Knockout or inhibition of CTSB decreased HUA-induced increased renal uric acid excretion in vivo

Next, we investigated the functional role of CTSB in HUA. First, conditional renal proximal tubular epithelial cell CTSB knockout (CTSB$^{tecKO}$) mice were generated; specifically, CTSB$^{flox/flox}$ mice were bred with ggt1-cre$^+$ mice to delete CTSB in renal proximal tubular epithelial cells (Fig. 2a). CTSB$^{flox/flox}$ mice without ggt1-cre recombinase were used as WT controls in all experiments. Analysis of protein expression showed that the CTSB knockout was nearly complete (Fig. 2b). Then, we assessed the SUA of CTSB$^{flox/flox}$ and CTSB$^{tecKO}$ mice in HUA. The results showed that the average SUA level of CTSB$^{tecKO}$ + HUA mice was lower than that of CTSB$^{flox/flox}$ + HUA mice (Fig. 2c). Compared with CTSB$^{flox/flox}$ mice, the average FEua in CTSB$^{flox/flox}$ + HUA mice were significantly reduced, and knockout of CTSB in renal proximal tubular epithelial cells increased the FEua (Fig. 2d). There were no differences in Scr or U-MALB among the groups (Fig. 2e, f). The results of serum IL-1β, Serum IL-6 and Serum TNF-α in HUA mice were increased, and CTSB knockout in renal proximal tubule epithelial cells reduced them (Fig. 2g–i). In addition, HUA upregulated URAT1 and GLUT9 expression and downregulated ABCG2 in the kidney. Interestingly, in CTSB$^{tecKO}$ mice, URAT1 and GLUT9 expression were attenuated, while ABCG2 expression was elevated (Fig. 2j–m), providing further evidence of reduced kidney uric acid reabsorption in CTSB$^{tecKO}$ mice. Similarly, there were no differences in HE staining of kidneys in among the groups (Fig. 2n).

We then used CA-074 methyl ester (CA074Me) to inhibit CTSB activity in mice. As in the prior results, after inhibiting CTSB, SUA of HUA mice was also decreased (Supplementary Fig. S3a). The FEua of the HUA mice also increased after CA074Me treatment (Supplementary Fig. S3b) but the increase did not reach statistical significance. There were no significant differences in Scr or U-MALB levels among the four groups of mice (Supplementary Fig. S3c, d). Serum IL-1β in mice also showed effects similar to CTSB knockout (Supplementary Fig. S3e). Finally, total kidney proteins were extracted from the four groups of mice to evaluate the expression of URAT1, ABCG2 and GLUT9. These results also indicate that CTSB inhibition led to decreased expression of renal URAT1 and GLUT9, and increased expression of ABCG2 (Supplementary Fig. S3f). There were no significant differences in HE staining among the kidneys of the four groups (Supplementary Fig. S3g).

### Knockout or inhibition of CTSB reversed the aberrant expression of urate transporters in vitro

To verify the in vivo results in vitro, a CTSB knockout (CTSB-KO) HK-2 cell line was constructed (Supplementary Fig. S4a). The qRT-PCR results showed that the CTSB knockout was effective (Supplementary Fig. S4b). Uric acid stimulation increased IL-1β in renal proximal tubular epithelial cells, while knockout of CTSB decreased IL-1β induced by uric acid (Supplementary Fig. S4c). URAT1 protein was significantly upregulated after uric acid stimulation, but after CTSB knockout, the level of URAT1 protein decreased (Supplementary Fig. S4d, e). Finally, CA074Me was used to treat HK-2 cells. Consistent with the above results, inhibition of CTSB reduced the abnormally increased IL-1β and URAT1 after uric acid stimulation (Supplementary Fig. S4f–h).

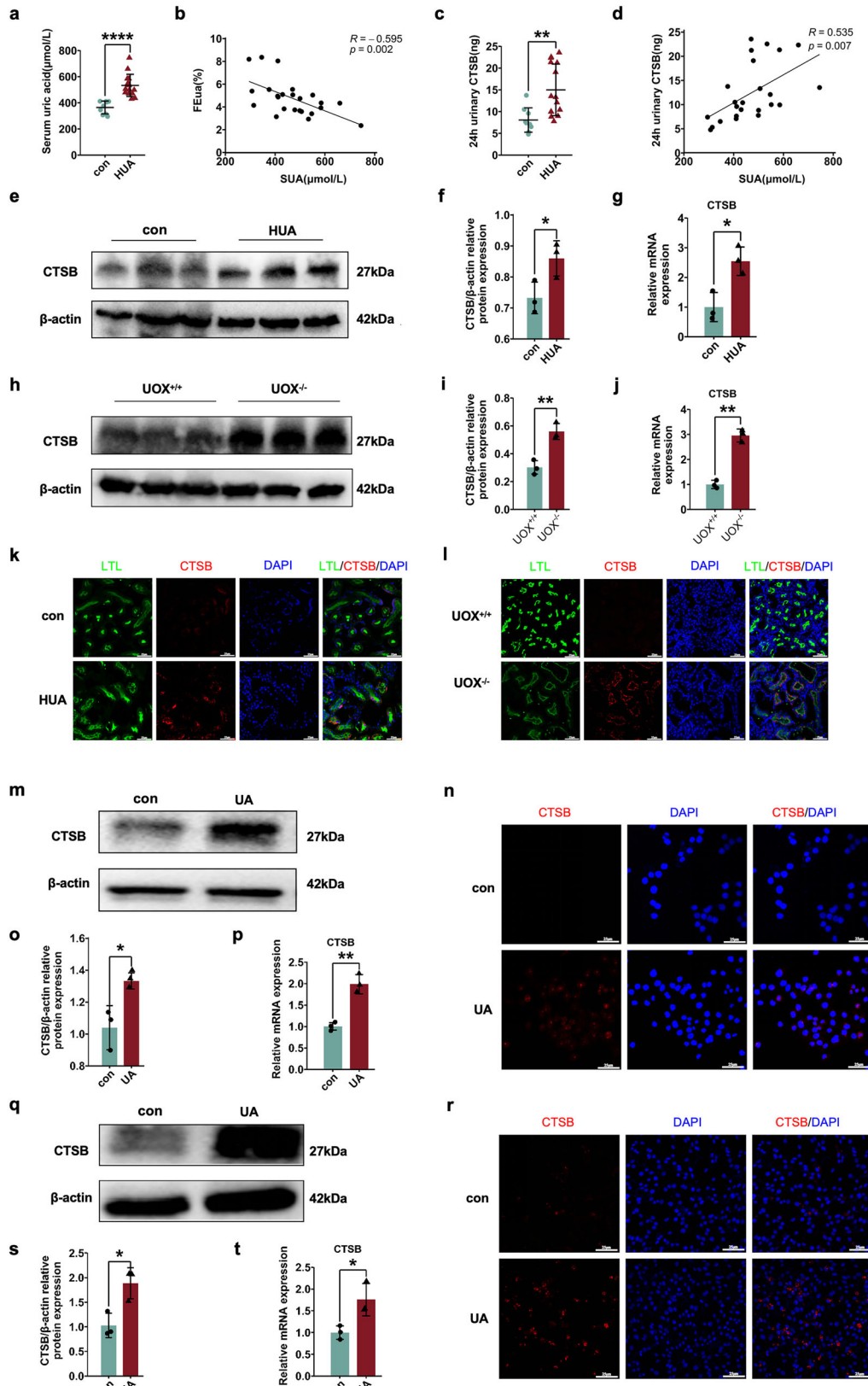

## CTSB knockout in renal proximal tubular epithelial cells changed renal metabolism

To further investigate the potential mechanisms associated with renal proximal tubular epithelial cell knockout of CTSB, kidneys of $CTSB^{tecKO} + HUA$ and $CTSB^{flox/flox} + HUA$ mice were selected for RNA-sequencing (n = 4). After quality control of the data, there were 208 differentially expressed genes in the kidneys of the two groups of mice, of which 124 were upregulated and 84 were downregulated (Fig. 3a). The top 10 differentially expressed genes in the kidneys of the two groups of mice are displayed in the volcano plot and heatmap (Fig. 3b, c). After knockout of

**Fig. 1 | CTSB accumulation in the urine of HUA patients, kidneys of HUA mice and proximal tubular epithelial cells. a** The SUA levels of the control (n = 9) and HUA (n = 15) patients. **b** Spearman analysis between SUA and FEua in patients. **c** Total CTSB in 24-hour urine of control and HUA patients. **d** Spearman analysis between SUA and 24-hour urinary CTSB in patients. **e** Western blot of CTSB in the kidneys of HUA mice. **f** Statistical analysis of CTSB protein expression in HUA mice. **g** mRNA expression levels of CTSB in the kidneys of HUA mice. **h** Western blot of CTSB in the kidneys of uricase knockout HUA mice. **i** Statistical analysis of CTSB expression in uricase knockout HUA mice. **j** CTSB mRNA expression in the kidneys of uricase knockout HUA mice. **k** Immunofluorescence staining of CTSB in the kidneys of HUA mice. **l** Immunofluorescence staining of CTSB in the kidneys of uricase knockout HUA mice. **m** Western blot of CTSB in HK-2 cells after uric acid stimulation. **o** Statistical analysis of CTSB protein expression. **p** CTSB mRNA levels in HK-2 cells after uric acid stimulation. **n** Immunofluorescence staining for CTSB after treatment with uric acid in HK-2 cells. **q** Western blot of CTSB in MRTECs after uric acid stimulation. **s** Statistical analysis of protein expression. **t** CTSB mRNA levels in MRTEC cells after uric acid stimulation. **r** Immunofluorescence staining of CTSB in MRTEC cells after treatment with uric acid. Bars = 25 μm. n ≥ 3. *$P < 0.05$, **$P < 0.01$, ***$P < 0.001$, ****$P < 0.0001$.

CTSB in renal proximal tubular epithelial cells of HUA mice, the top 10 upregulated genes were RIKEN cDNA A730046G19 gene (A730046J19Rik), wiskott-Aldrich syndrome protein homolog (LOC118568183), early growth response 1 (Egr1), G protein-coupled receptor 1 (Gpr1), yippee like 4 (Ypel4), cytochrome P450 family 11 subfamily A member 1 (Cyp11a1), cytochrome P450 family 11 subfamily B member 1 (Cyp11b1), cytochrome P450 family 21 subfamily A member 1 (Cyp21a1), IQ motif and ubiquitin domain containing (Iqub), and ATPase H + /K+ transporting subunit beta (Atp4b). The top 10 downregulated genes were calpain 11 (Capn11), transcription factor 23 (Tcf23), ankyrin repeat and death domain containing 1 A (Ankdd1a), signaling lymphocytic activation molecule family member 1 (Slamf1), protocadherin alpha 12 (Pcdha12), killer cell lectin like receptor C1 (KlRC1), solute carrier family 16 member 8 (Slc16a8), double homeobox B-like 1 (Duxbl1), myeloperoxidase (Mpo), and dihydropyrimidinase like 4 (Dpysl4). Next, these differentially expressed genes were subjected to GO and KEGG enrichment analyses. The results of GO and KEGG analyses showed that many pathways changed in HUA mice after CTSB knockout in renal proximal epithelial cells (Fig. 3d–g). For example, during HUA, compared with CTSB^flox/flox mouse kidneys, multiple pathways, such as glycolysis and the NOD-like receptor signaling pathway, were downregulated in CTSB^tecKO mouse kidneys, while multiple pathways, such as regulation of T-cell differentiation and cytokine–cytokine receptor interaction, were upregulated.

To verify whether CTSB affected the metabolism of mouse kidneys, nontargeted metabolomic analysis was conducted on the kidneys of CTSB^flox/flox and CTSB^tecKO mice. Metabolomics detected various metabolites, including fatty acyls, glycerol phospholipids, steroids, and their derivatives (Supplementary Fig. S5a). The kidneys of the two groups of mice differentially expressed multiple metabolites, and a volcano plot was constructed to visualize them (Supplementary Fig. S5b). Then, based on the metabolomics results, a heatmap of the differentially expressed metabolites of TOP50 was constructed (Supplementary Fig. S5c). The results showed that compared with kidneys of CTSB^flox/flox mice, multiple metabolites in the kidneys of CTSB^tecKO mice were downregulated, and some of the down-regulated metabolites were involved in glycolysis (e.g., D-glutamine). Finally, we performed KEGG enrichment analysis of differentially expressed metabolites, and the results showed that multiple pathways were upregulated in the kidneys of CTSB^tecKO mice (Supplementary Fig. S5d).

### Glycolysis of renal proximal tubular epithelial cells was increased in vivo and in vitro during HUA

As the key enzymes of the glycolysis pathway, type 2 hexokinase (HK2) and type M2 pyruvate kinase (PKM2) in the kidneys of control and HUA mice were evaluated. At the protein level, the expression of HK2 and PKM2 in mouse kidneys was significantly upregulated during HUA (Fig. 4a–c). In addition, the immunofluorescence results from the mice kidneys also showed an increase in HK2 and PKM2 (Fig. 4d, e). In addition, an increase in HK2 and PKM2 was also observed in the kidneys of UOX ^−/− mice (Fig. 4f–j).

In HK-2 cells, the expression of HK2 and PKM2 significantly increased after intervention with uric acid (Fig. 5a–c). Similarly, similar results were found in the MRTECs, with significantly upregulated expression of HK2 and PKM2 after stimulation with uric acid (Fig. 5d–f). Finally, the activities of hexokinase and pyruvate kinase and the levels of lactate in HK-2 cells after

uric acid stimulation were measured. After treatment with uric acid, hex-okinase activity, pyruvate kinase activity, and lactate levels were significantly increased (Fig. 5g–i).

### CTSB knockout or inhibition reduced enhanced glycolysis induced by HUA in vivo and in vitro

Consistent with the results of RNA-sequencing, the immunofluorescence results showed that CTSB knockout in renal proximal tubular epithelial cells led to downregulated expression of HK2 and PKM2 (Fig. 6a, b). In HUA mice, we also found that CTSB knockout downregulated the expression of HK2 and PKM2 in the kidney (Fig. 6c). Finally, in renal proximal tubular epithelial cells, we found that CTSB knockout reversed UA-induced upregulation of HK2 and PKM2 expression after uric acid treatment (Fig. 6d–f). We then used CA074Me to inhibit CTSB in vivo and in vitro. Protein expression of HK2 and PKM2 was downregulated after CTSB inhibition in the kidneys of HUA mice (Supplementary Fig. S6a). Protein expression of HK2 and PKM2 was also decreased after CTSB was inhibited in vitro (Supplementary Fig. S6b–d).

### Inhibition of glycolysis promoted renal uric acid excretion

To verify the role of glycolysis in HUA, the HK2 inhibitor lonidamine (Lo) was used to inhibit glycolysis both in vivo and in vitro. Serum levels of IL-1β, IL-6, TNF-α, as well as URAT1, GLUT9, ABCG2, FEua, and SUA were determined.

In mice, HK2 inhibition led to a decrease in SUA (Fig. 7a) and an increase in FEua (Fig. 7b), with no significant changes in Scr or U-MALB (Fig. 7c, d). Serum IL-1β, IL-6, and TNF-α levels were reduced (Fig. 7e–g). Additionally, Renal expression of HK2, PKM2, URAT1, and GLUT9 were decreased, while ABCG2 was increased after HK2 inhibition (Fig. 7h–m). In vitro, glycolysis inhibition reversed the HUA-induced upregulation of IL-1β and URAT1 (Fig. 7n–p).

### CTSB overexpression promoted glycolysis and upregulated the expression of URAT1 and GLUT9, while reducing ABCG2 expression

We then assessed overexpression of CTSB in renal proximal tubular epithelial cells to validate the aforementioned findings. Firstly, HK-2 cells overexpressing CTSB were constructed. The CTSB-OE group exhibited a significant elevation in CTSB levels compared to the CTSB-NC group (Fig. 8a, b). Next, the expression of URAT1, GLUT9 and ABCG2 in cells were determined after overexpression of CTSB. Overexpression of CTSB led to an upregulation of URAT1 and GLUT9 protein levels and a down-regulation of ABCG2 expression. Notably, these aberrant expression patterns were partially reversed following treatment with the CA074Me (Fig. 8c–f). Finally, the impact of CTSB overexpression on glycolysis was investigated. As expected, Overexpression of CTSB led to an upregulation of HK2 and PKM2 protein expression, which was partially reversed by treatment with the CA074Me (Fig. 8g–i). Furthermore, ECAR assays revealed that glycolysis in proximal tubular cells was significantly elevated under conditions of uric acid stimulation and CTSB overexpression (Fig. 8j).

### Discussion

In this study, we observed the upregulation of CTSB in renal proximal tubular epithelial cells during high uric acid stimulation. In vivo, compared

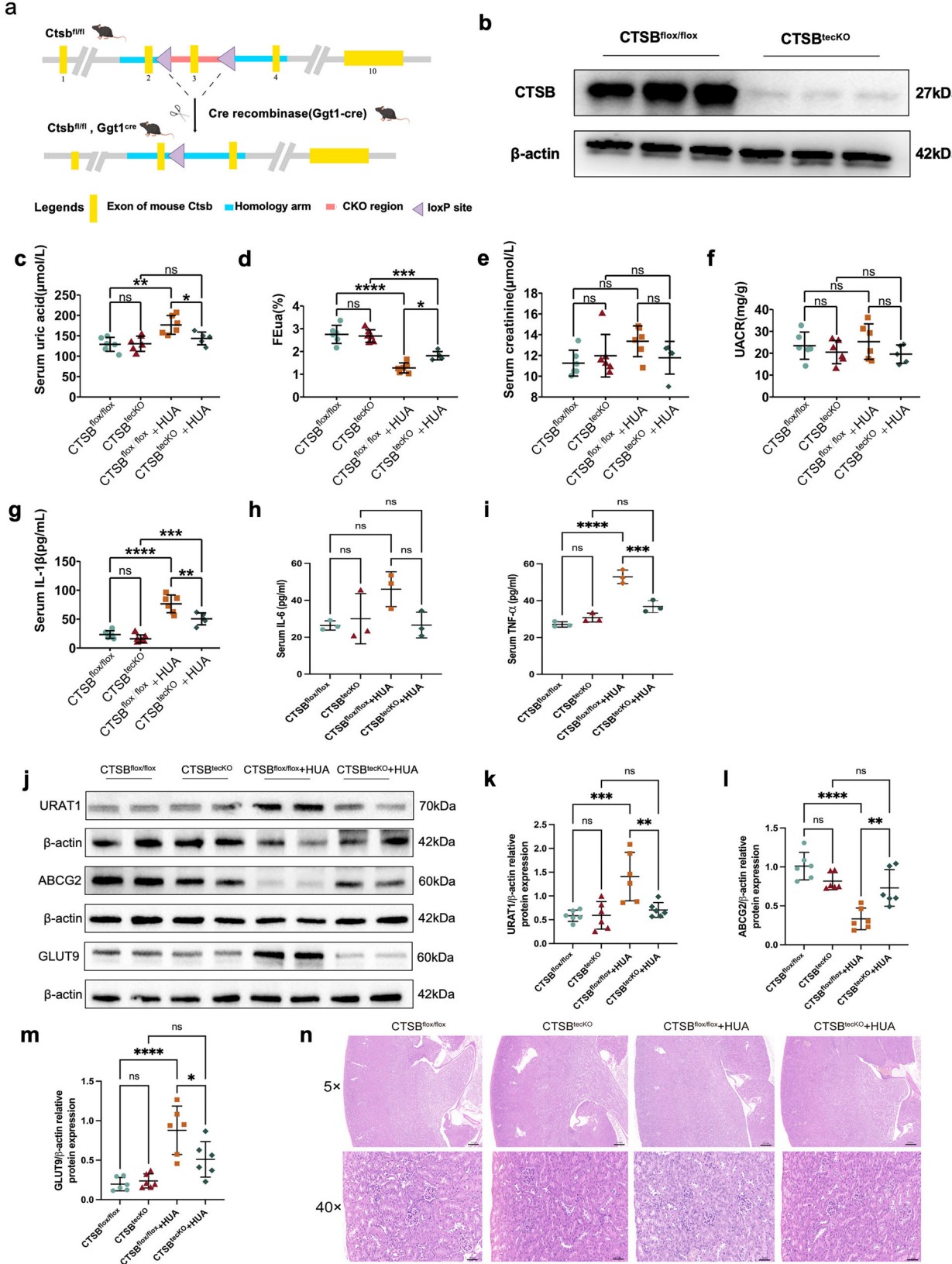

**Fig. 2 | CTSB knockout in renal proximal tubular epithelial cells increased renal uric acid excretion. a** Schematic strategy for the generation of CTSB^tecKO mice. **b** Verification of CTSB in CTSB^tecKO mice by western blot. **c** The levels of SUA, **d** FEua, **e** Scr, **f** U-MALB, **g** Serum IL-1β, **h** Serum IL-6 and **i** Serum TNF-α in different groups of mice. **j** Western blot of URAT1, ABCG2, and GLUT9 in the kidneys of different groups of mice. Statistical analysis of **k** URAT1, **l** ABCG2, and **m** GLUT9 protein expression. **n** HE staining of kidneys of different groups of mice. SUA serum uric acid, FEua fractional excretion of uric acid, Scr serum creatinine, U-MALB urine microalbumin. Bars = 25 μm (40×). Bars = 200 μm (5×). n ≥ 3. *P < 0.05, **P < 0.01, ***P < 0.001, ****P < 0.0001.

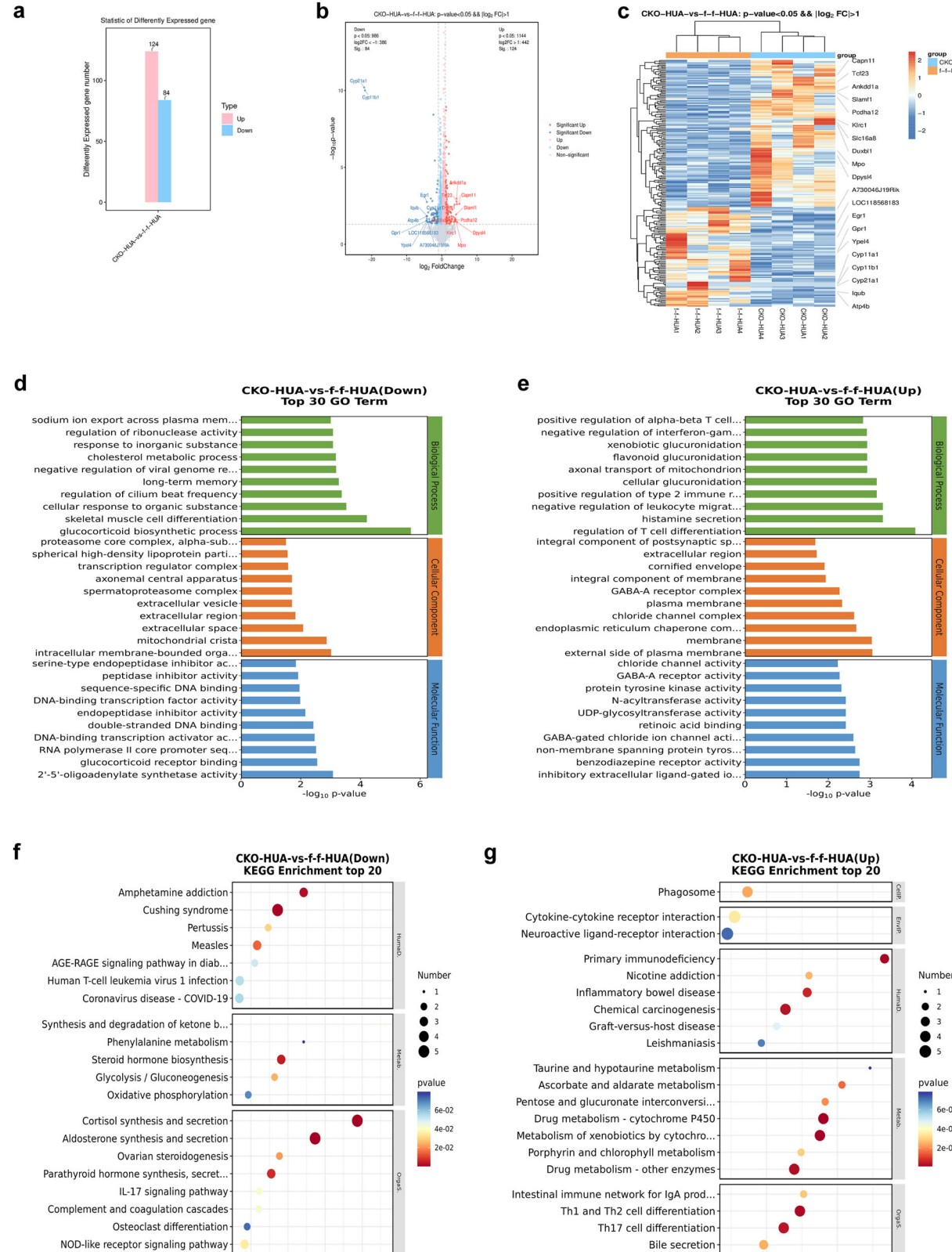

**Fig. 3 | RNA-sequencing showed that CTSB knockout in renal proximal tubular epithelial cells decreased glycolysis. a** DEG after CTSB knockout in the renal proximal tubular epithelial cells of HUA mice. **b** Volcano plot and **c** heatmap of the top DEGs in renal proximal tubular epithelial cells in HUA mice after CTSB knockout. **d** GO analysis of downregulated and **e** upregulated pathways after CTSB knockout in the renal proximal tubular epithelial cells of HUA mice. **f** KEGG analysis of downregulated and **g** upregulated pathways in renal proximal tubular epithelial cells after CTSB knockout in HUA mice. n = 4.

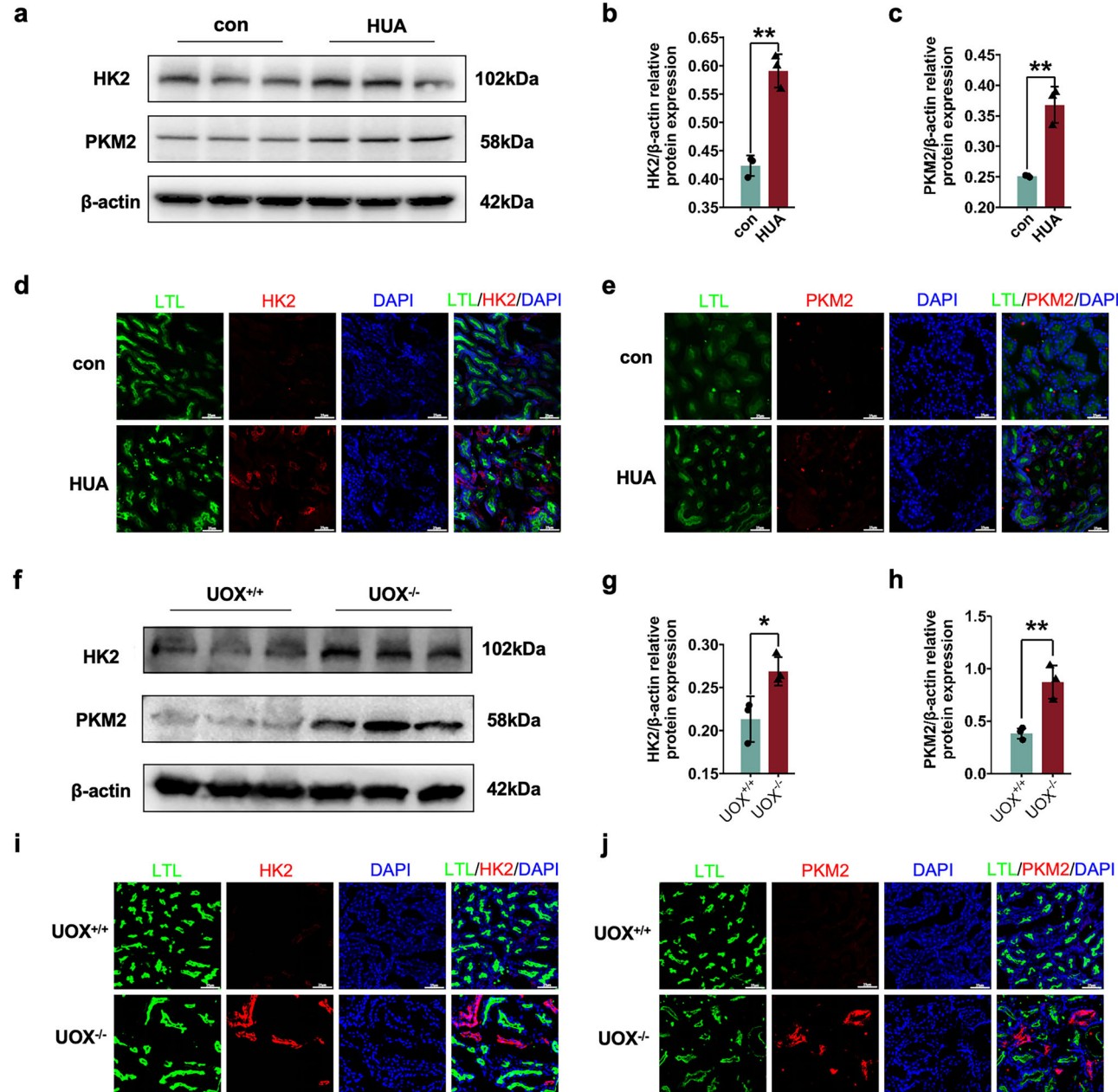

**Fig. 4 | Glycolysis was increased in the kidneys of HUA mice. a** Western blot of HK2 and PKM2 in kidneys of HUA mice. Statistical analysis of **b** HK2 and **c** PKM2 protein expression. Immunofluorescence staining of **d** HK2 and **e** PKM2 in the kidneys of HUA mice. **f** Western blot of HK2 and PKM2 in the kidneys of uricase-knockout mice. Statistical analysis of **g** HK2 and **h** PKM2 protein expression. Immunofluorescence staining of **i** HK2 and **j** PKM2 in the kidneys of uricase knockout mice. Bars = 25 μm. n ≥ 3. *$P < 0.05$, **$P < 0.01$.

with CTSB$^{flox/flox}$ + HUA mice, CTSB$^{tecKO}$ + HUA exhibited downregulated URAT1, GLUT9, IL-1β, IL-6 and TNF-α, upregulated ABCG2, increased FEua, and decreased SUA. In vitro, CTSB knockout reversed the alterations in urate transporter expression induced by high uric acid exposure. Similarly, pharmacological inhibition of CTSB using CA074Me produced comparable effects. Moreover, glycolysis inhibition promoted renal uric acid excretion both in vivo and in vitro. Finally, CTSB overexpression enhanced glycolysis, resulting in increased expression of URAT1 and GLUT9, and decreased ABCG2 expression.

Primary HUA is mostly caused by reduced renal uric acid excretion[6], but the mechanism is still not clear. Most HUA mice were not examined for renal uric acid excretion markers[18–20], or drug treatments caused increased Scr[21–23], in which serious kidney injury masks the effects of HUA and is unable to represent the clinical scenario of HUA without chronic kidney

disease. In this study, we successfully constructed an HUA mouse model with elevated SUA, reduced FEua, and normal Scr and U-MALB to simulate simple HUA patients without obvious renal insufficiency, which could rule out factors related to reduced glomerular uric acid filtration caused by renal insufficiency and enabling improved analysis of renal tubular uric acid handling disorders during HUA. Since recent studies have shown that the incidence of hyperuricemia is higher in males than in females, we focused only on males in this study[24].

Uric acid metabolism is complex, involving uric acid production in the liver and renal and intestinal excretion[25], among which renal uric acid excretion accounts for 2/3 of total body excretion. The excretion of uric acid from the blood to the kidney is a complicated process. First, almost all uric acid filters through the glomeruli. Subsequently, almost all (99%) of the filtered uric acid is reabsorbed in the proximal tubule through the action of

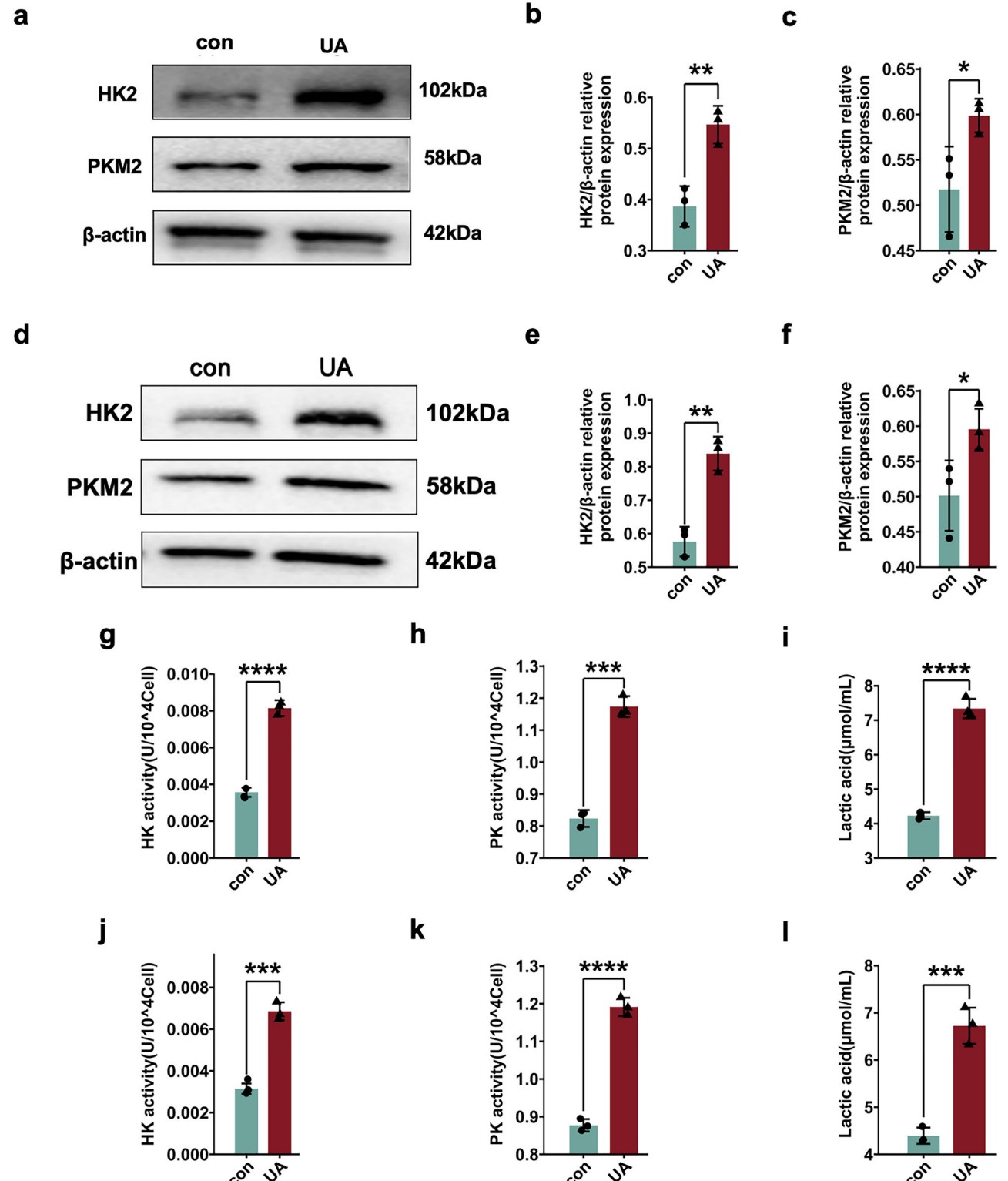

**Fig. 5 | Glycolysis was enhanced after uric acid stimulation in renal proximal tubular epithelial cells. a** Western blot of HK2 and PKM2 in HK-2 cells after uric acid stimulation. Statistical analysis of **b** HK2 and **c** PKM2 protein expression HK-2 cells after uric acid stimulation. **d** Western blot of HK2 and PKM2 in MRTEC cells after uric acid stimulation. Statistical analysis of **e** HK2 and **f** PKM2 protein expression in MRTECs after uric acid stimulation. **g** Hexokinase activity, **h** pyruvate kinase activity, and **i** lactate levels after uric acid stimulation in HK-2 cells. **j** Hexokinase activity, **k** pyruvate kinase activity, and **l** lactate levels after uric acid stimulation in MRTEC cells. *$P < 0.05$, **$P < 0.01$, ***$P < 0.001$, ****$P < 0.0001$.

multiple uric acid reabsorption proteins. Thereafter, approximately 50% of the reabsorbed uric acid is secreted back into the tubules, and then approximately 40% of the uric acid can be reabsorbed into the blood through the uric acid reabsorption protein at the distal end of the renal proximal tubules. Ultimately, only ~10% of filtered uric acid is excreted in urine, a process tightly regulated by multiple transporters[26,27]. These complex uric acid processes in the kidney are accomplished by the synergistic action of multiple uric acid transporters, the most important of which is

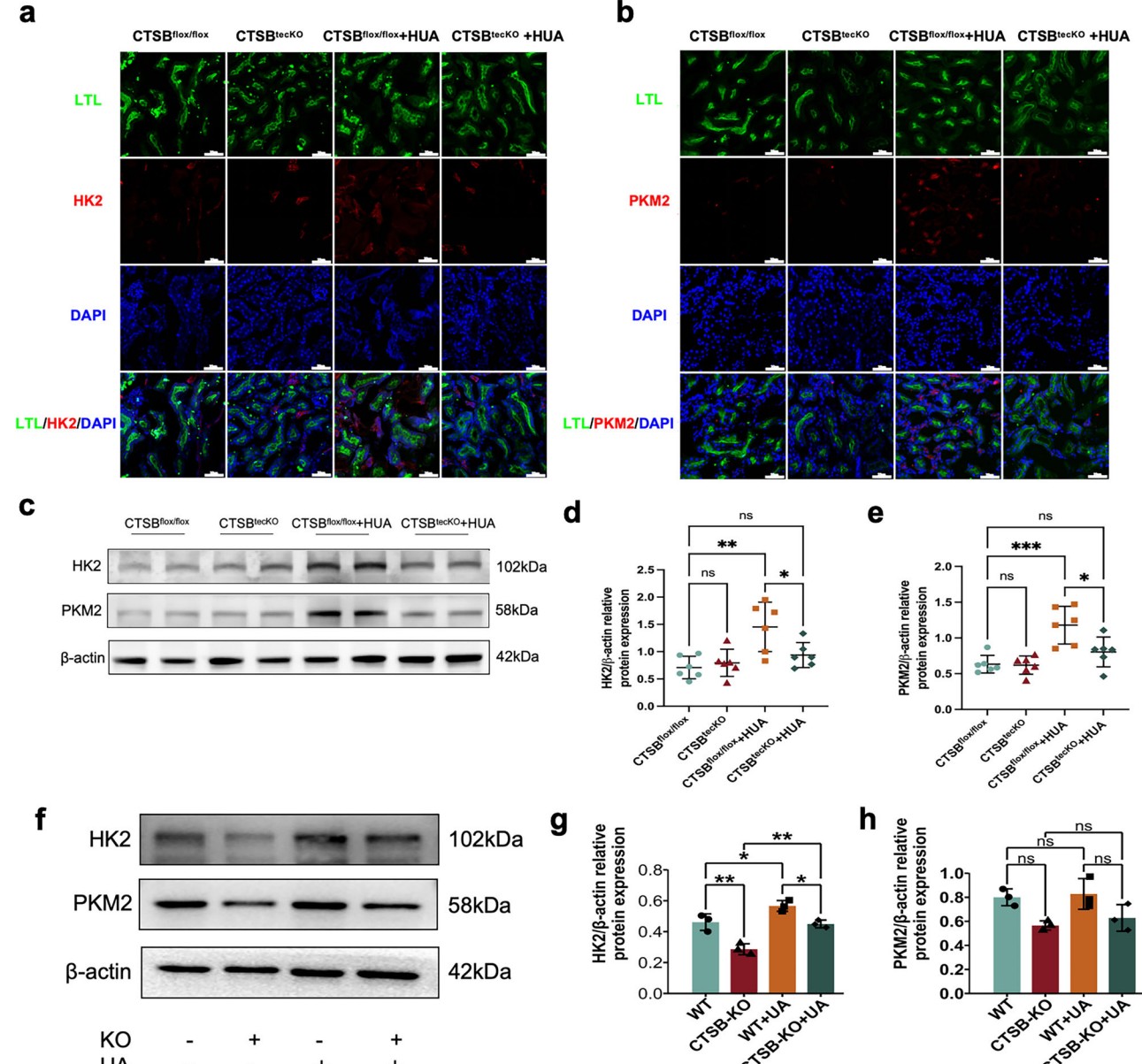

**Fig. 6 | CTSB knockout reduced HUA-induced glycolysis both in vivo and in vitro.** Immunofluorescence of **a** HK2 and **b** PKM2 in the kidneys of different groups of mice. **c** Western blot of HK2 and PKM2 in different groups of mice. Statistical analysis of **d** HK2 and **e** PKM2 protein expression. **f** Western blot of HK2 and PKM2 in CTSB-knockout renal proximal tubular epithelial cells after uric acid stimulation. Statistical analysis of **g** HK2 and **h** PKM2 protein expression in CTSB-knockout renal proximal tubular epithelial cells after uric acid stimulation. Bars = 25 μm. n ≥ 3. *$P < 0.05$.

URAT1[8]. In fact, numerous studies have found that URAT1 was significantly upregulated in the kidneys during HUA[28,29], suggesting increased renal uric acid reabsorption and decreased excretion. It has recently been reported that the selective URAT1 inhibitor, dotinurad, ameliorates the metabolic parameters and renal function in HUA patients, increases energy expenditure, and decreases lipid synthesis and inflammation in the liver and adipose tissue of rats, suggesting that URAT1 is connected to energy metabolism and inflammation[30]. Currently, there are drugs that target URAT1 to lower uric acid, such as benzbromarone. However, once drug usage is discontinued, SUA returns. Therefore, it is necessary to further study the mechanisms of abnormal renal uric acid excretion during HUA.

There are 20 known subtypes of cathepsins, 11 subtypes of which are present in humans. Cathepsins are expressed in many parts of the kidney; cathepsin B, D, and L are mainly expressed in podocytes; cathepsin B, D, and S are mainly expressed in endothelial cells; and cathepsin B, D, G, and L are mainly expressed in renal tubule cells, among which CTSB has been the

most studied[15]. Cathepsins are a kind of protease first found in the gastric mucosa, which are active in a slightly acidic environment. Several studies have shown that cathepsins are involved in the occurrence and/or development of a variety of kidney diseases, including acute kidney injury, chronic kidney disease, and diabetic nephropathy[15]. CTSB, originally named cathepsin B1, is the first and most characteristic member of the papain-like lysosome cysteine peptidase C1 family, which was first purified from human liver in 1973 and is commonly expressed in most cell and tissue types[31]. In recent decades, it has been found that CTSB is involved in the occurrence and/or development of a variety of diseases and plays a crucial role in them. For example, in mouse and cell experiments, it was found that CTSB knockout or inhibition can improve the progression of a variety of diseases, such as acute myelogenous leukemia, lung cancer, pancreatitis, a variety of neurological diseases and myocardial injury[32–35]. In the kidney, multiple studies have shown that CTSB is also involved in a variety of kidney diseases[36–38]. In mice, the use of CTSB-specific inhibitors can alleviate

**Fig. 7 | Inhibition of glycolysis increased renal uric acid excretion.** The levels of **a** SUA, **b** FEua, **c** Scr, **d** U-MALB, **e** Serum IL-1β, **f** Serum IL-6 and **g** Serum TNF-α in different groups of mice. **h** Western blot of HK2, PKM2, URAT1, ABCG2 and GLUT9 after inhibition of glycolysis in vitro. Statistical analysis of **i** HK2, **j** PKM2, **k** URAT1, **l** ABCG2 and **m** GLUT9 protein expression. **n** IL-1β levels after inhibition of glycolysis in vitro. **o** Western blot of URAT1 after inhibition of glycolysis in vitro. **p** Statistical analysis of URAT1 protein expression. n ≥ 3. *$P < 0.05$, **$P < 0.01$, ***$P < 0.001$, ****$P < 0.0001$.

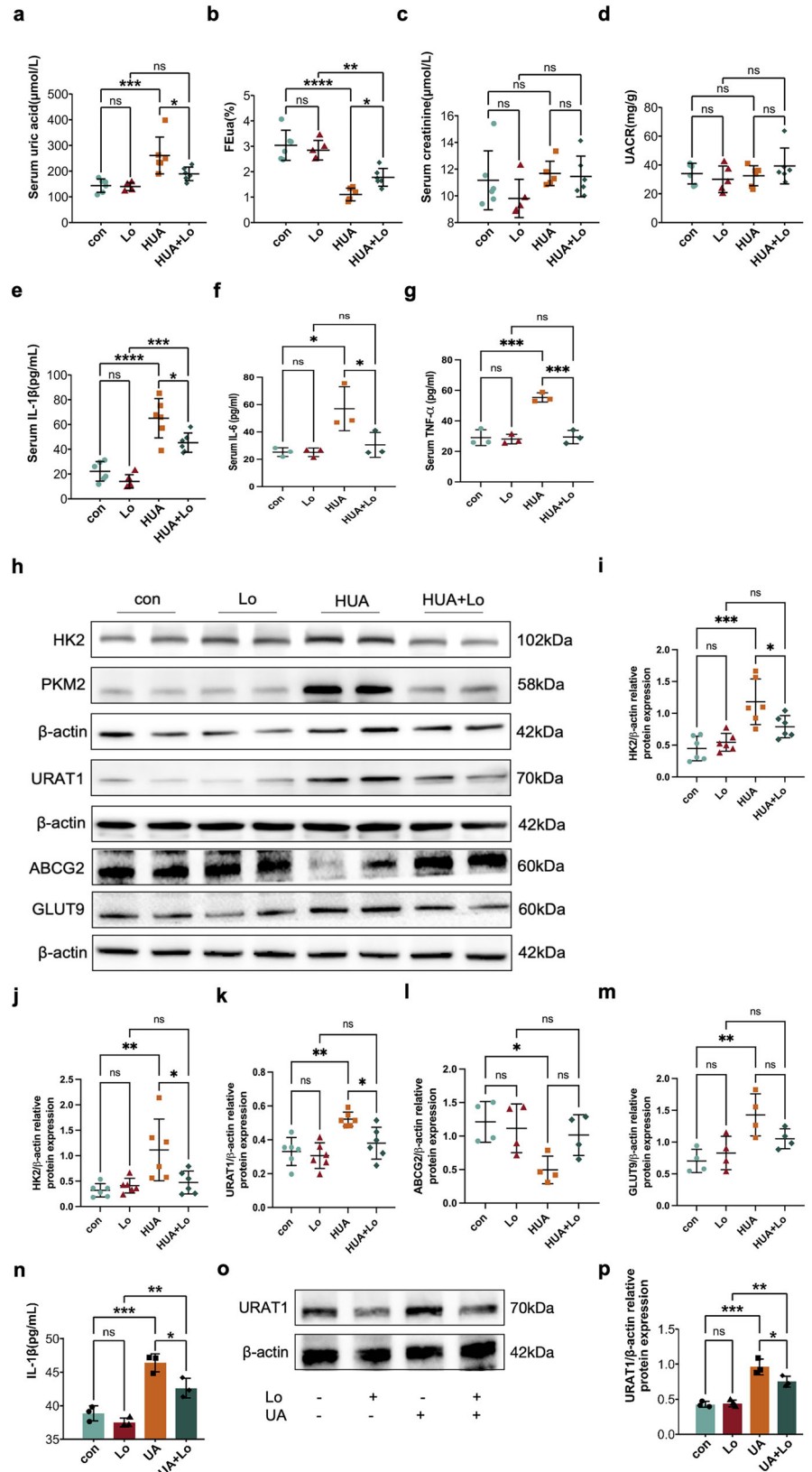

lipopolysaccharide (LPS)-induced mitochondrial damage in renal tubular epithelial cells[39]. In addition, in renal ischemia–reperfusion injury mice, CTSB and CTSD were released from lysosomes into the cytoplasm, activating NLRP3, which in turn promoted inflammation[40]. However, there is no study addressing the role of CTSB in reduced renal uric acid excretion during HUA. Our study revealed the involvement of CTSB in inflammation

and reduced renal uric acid excretion during HUA, suggesting that CTSB intervention may be a new target for the treatment of HUA.

Studies have reported that the mechanisms of CTSB involvement in disease mainly involve degradation of the extracellular matrix, activation of inflammation, and participation in autophagy[14,41,42]. Recent studies have also found that CTSB plays a significant role in diseases by participating in

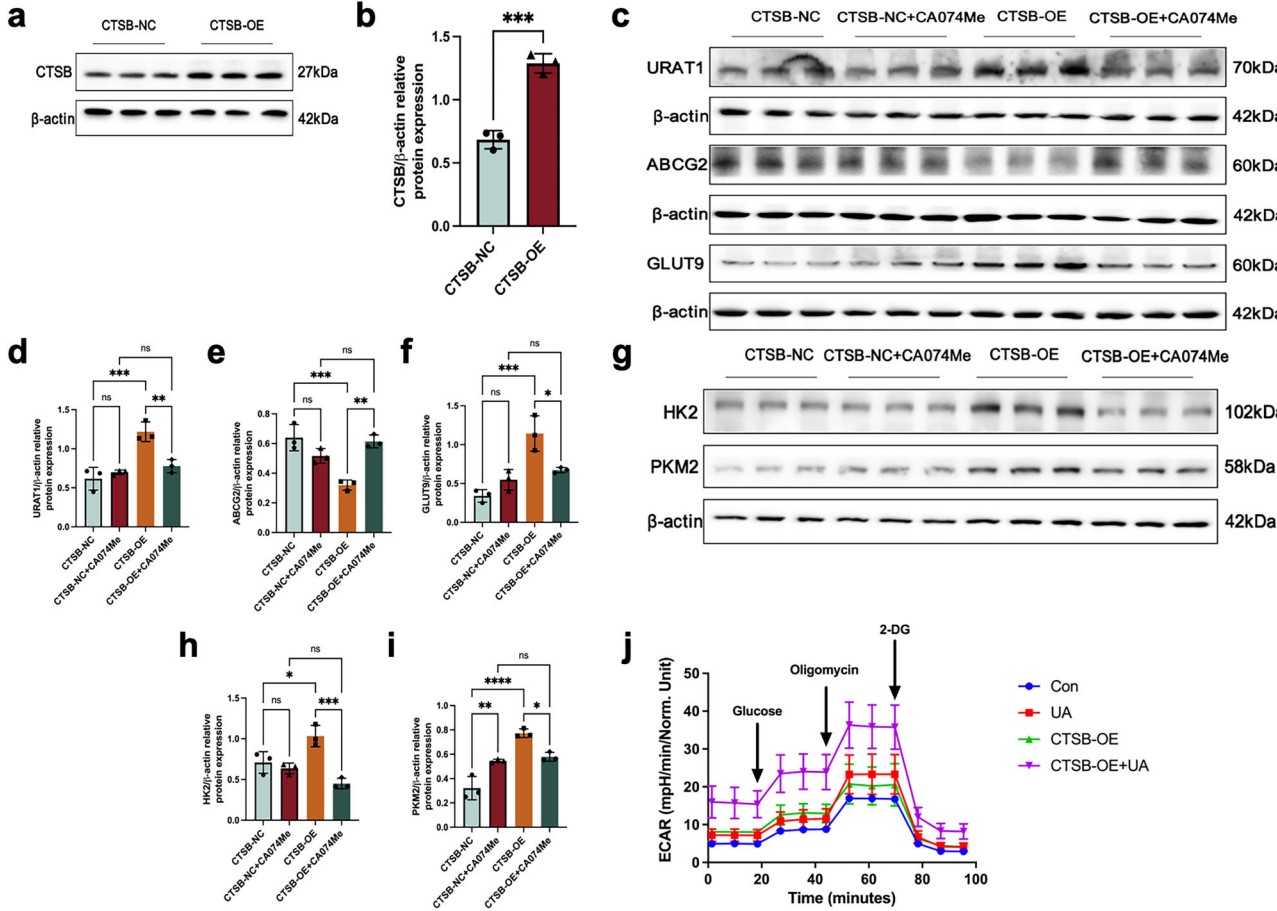

**Fig. 8 | CTSB overexpression promoted glycolysis and upregulated urate reabsorption transporters in vitro. a** Western blot of CTSB after CTSB overexpression in HK-2 cells. **b** Statistical analysis of CTSB protein expression. **c** Western blot of URAT1, ABCG2 and GLUT9 protein expression. Statistical analysis of **d** URAT1, **e** ABCG2 and **f** GLUT9 protein expression. **g** Western blot of HK2 and PKM2 after CTSB overexpression in HK-2 cells. Statistical analysis of **h** HK2 and **i** PKM2 protein expression. **j** ECAR results after CTSB overexpression in HK-2 cells. n = 3. *$P < 0.05$, **$P < 0.01$, ***$P < 0.001$, ****$P < 0.0001$.

glycolysis[43,44]. Some researchers found that monosodium urate (MSU) mediated inflammation in macrophages by promoting glucose transporter 1 (GLUT1)-mediated glycolysis[45]. However, the effects of CTSB on glycolysis in renal proximal tubular epithelial cells in HUA have not been studied. Here, we found enhancement of glycolysis in renal proximal tubular epithelial cells in vivo and in vitro, as evidenced by the upregulated expression and activity of key glycolytic enzymes such as HK2 and PKM2. After the construction of HUA in CTSB[flox/flox] and CTSB[tecKO] mice, transcriptomic sequencing results showed that the glycolysis pathway was downregulated, which was consistent with our RNA-sequencing results. We also observed that glycolysis was significantly inhibited in the kidneys of CTSB[tecKO] HUA mice. These results suggested that CTSB could promote the occurrence of glycolysis in the kidney during HUA, which could be alleviated after CTSB knockout in renal proximal tubular epithelial cells. The results in vitro are also consistent with these findings. Therefore, we conclude that CTSB may be involved in the inflammation and decreased renal uric acid excretion in the kidneys of HUA through glycolysis.

Our RNA-sequencing results revealed that *Dpysl4* expression was reduced in CTSB-deficient mice. Previous studies have reported that DPYSL4 acts as a negative regulator of glycolysis by suppressing glycolytic enzyme expression and glucose uptake[46], which is consistent with our findings and supports the inhibitory effect of CTSB knockout on glycolytic activity. In addition, we observed decreased expression of *Egr1* following CTSB deletion. Given that EGR1 has been shown to promote glycolysis in various contexts[47–49], our results further suggest a potential role for CTSB in modulating glycolysis via regulation of EGR1 expression. However, it is worth noting that in hepatocellular carcinoma, EGR1 has been reported to

suppress glycolysis[50], indicating that the functional role of EGR1 may vary depending on disease context or tissue type. These observations highlight the complexity of glycolysis regulation and suggest that CTSB may influence glycolysis in part through EGR1 and DPYSL4 mediated pathways. Further experimental validation is needed to confirm the functional relevance of DPYSL4 and EGR1 in CTSB-mediated glycolytic regulation in renal tubules. Although our metabolomics results did not directly indicate changes in glycolysis, an enhancement of the tricarboxylic acid (TCA) cycle was observed, and several studies have found that glycolysis decreased when the TCA cycle was upregulated[51,52]. These results suggest that after the knockout of CTSB in mouse renal proximal tubular epithelial cells, there were significant changes in mouse kidney metabolism, including an increase in the TCA cycle and a decrease in glycolysis.

The kidney is one of the most active organs in the body with respect to energy metabolism[53]. As the main site of substance transport in the kidney, the renal proximal tubule plays an important role in kidney metabolism. There are relatively few studies on the role of glycolysis in HUA, and it has been reported that after uric acid stimulation in mesenchymal stem cells and macrophages in vitro, glycolysis was activated, thus inducing a downstream inflammatory response[45,54]. Here, we demonstrated in vivo and in vitro that HUA upregulated glycolysis in renal proximal tubular epithelial cells. However, the means by which HUA induces renal tubule glycolysis needs further study. A previous study found that in HUA rats and renal proximal tubular epithelial cells, HUA led to an increase in NOD-like receptor thermal protein domain associated protein 3 (NLRP3)[13,55]. This result has also been confirmed in other studies[56–58]. Interestingly, it has been suggested that CTSB can also promote NLRP3 activation[59], which promotes glycolysis,

and inhibition of NLRP3 has been shown to ameliorate abnormally increased glycolysis in mice[60–62]. NLRP3 can regulate glycolysis by increasing 6-phosphofructo-2-kinase (PFKFB3) in macrophages in an IL-1β-dependent manner[61]. Based on prior studies linking NLRP3 to glycolytic flux and urate handling, we hypothesize a CTSB-NLRP3-glycolysis axis, which causes reduced renal uric acid excretion. The RNA-sequencing results in CTSB[tecKO] mice also showed downregulation of the NLRP3 pathway in the kidney. How glycolysis causes reduced renal uric acid excretion is also a question worthy of further study. Thangaraju et al. found that, in mice with sodium/lactate transporter knockout, SUA levels decreased and urinary uric acid and urinary lactate increased[63]. Here, we speculated that during HUA, glycolysis in renal proximal tubular epithelial cells increases, and the levels of lactate, as one of the glycolysis products, increases. The increased lactate levels may lead to reduced renal uric acid reabsorption, leading to a compensatory increase in systemic inflammation and abnormal renal uric acid excretion, which aggravates the progression of HUA. It will be important and interesting to evaluate these hypotheses experimentally in future.

Despite the multilayered evidence provided in this study, several limitations should be acknowledged. First, our research focused exclusively on male mice and male patients, given the higher prevalence of hyperuricemia in males; however, potential sex-specific differences in CTSB expression and uric acid metabolism require further investigation. Second, although we identified that CTSB regulates glycolysis and subsequently alters urate transporter expression, the molecular mechanisms by which CTSB modulates glycolysis remain to be elucidated. The downstream effects of glycolysis on the regulation of key urate transporters—URAT1, GLUT9, and ABCG2—also need further investigation. Third, while this study highlights the CTSB–glycolysis–urate handling axis, its relevance in other pathological settings such as CKD, gout, or metabolic syndrome remains to be validated. Finally, although we proposed a role for NLRP3 and lactate as potential mediators linking glycolysis to inflammation and urate transport, these hypotheses were not directly tested and should be addressed in future studies.

In conclusion, our study demonstrates that activation of CTSB contributes to impaired renal uric acid excretion by regulating glycolysis. In vivo, CTSB knockout in renal proximal tubular epithelial cells reduces SUA levels by downregulating URAT1 and GLUT9, upregulating ABCG2, and enhancing renal uric acid excretion. Additionally, CTSB deletion alleviates systemic inflammation in HUA mice, as evidenced by decreased serum levels of IL-1β, IL-6, and TNF-α. In vitro, CTSB knockout reverses HUA-induced changes in urate transporters, while CTSB overexpression promotes glycolysis and further disrupts urate transporter expression. These findings highlight the CTSB-glycolysis axis as a critical regulator of urate homeostasis, providing a promising therapeutic target for hyperuricemia. Our findings provide new insights for the development of effective and safe CTSB-glycolysis inhibitors to correct abnormal urate handling in the renal tubules. Future studies should aim to validate these mechanisms in human kidney tissues and assess the potential of CTSB inhibitors in clinical settings.

## Methods
### Clinical and laboratory measurements
This study was approved by the Ethics Committee of Huadong Hospital affiliated to Fudan University (2022K186), and all participants provided signed informed consent. All procedures performed in studies involving human participants followed the Declaration of Helsinki 1964 and its later amendments or comparable ethical standards. Urine samples from 24 adult male patients were collected over 24 hours into a clean plastic bucket with a lid. In 24-hour urine collection, the first morning void is discarded and all urine is collected for the next 24 hours, including the first morning void the next day. During collection, the urine sample are stored in an environment of 2–8 °C or in a cool and ventilated place and mixed well after each urination.

When the 24-hour sample collection was completed, the sample was immediately delivered to the clinical laboratory for analysis of uric acid,

creatinine, sodium, potassium, glucose, albumin, and urine volume. Renal function and ability to handle uric acid were estimated by fractional excretion of uric acid (FFua). FEua was calculated as FEua = (Urinary uric acid (UUA) × Serum creatinine (Scr))/(SUA × Urinary creatinine (Ucr)) × 100%.

A 50 ml aliquot of the 24-hour urine sample was centrifuged at 500 g for 10 minutes, and 1.5 mL of the supernatant was removed. The 50 mL treated sample was frozen at −80 °C until analysis. CTSB in urine was detected using the human cathepsin B ELISA kit.

### Materials and reagents
Uric acid (UA) (#U2625) and potassium oxonate (PO) (#156124) were purchased from Sigma (St. Louis, MO, USA). CTSB inhibitor, CA-074 methyl ester (CA074Me) (#HY-100350), and glycolysis inhibitor, Lonidamine (Lo) (#HY-B0486), were obtained from MCE (New Jersey, USA). Anti-CTSB (#31718S), anti-rabbit IgG, HRP-linked antibody (#7074), and anti-mouse IgG, HRP-linked antibody (#7076), were purchased from CST (Boston, USA). Anti-HK2 (#66974-1-Ig), anti-PKM2 (#15822-1-AP), anti-URAT1 (#14937-1-AP), anti-ABCG2 (#27286-1-AP) and anti-GLUT9 (#67530-1-Ig) antibodies were purchased from Proteintech (Chicago, USA).

### Animals
All animal studies were approved by the Committee on the Ethics of Animal Experiments of Fudan University (2023-HDYY-74JZS). All experimental procedures were performed in compliance with animal protocols approved by the Institutional Animal Care and Use Committee at the National Institutes of Health and Fudan University. CTSB[flox/+] and ggt1cre[+] mice were obtained from Cyagen Biosciences (Suzhou, China). Renal proximal tubular-specific CTSB knockout mice (CTSB[tecKO]) were generated by cross-breeding CTSB[flox/flox] mice with ggt1cre[+] mice. Littermates with homozygous flox expression without Cre expression (CTSB[flox/flox]; ggt1cre[-]) were used as WT controls (CTSB[flox/flox]).

Genotypes were identified by polymerase chain reaction (PCR) amplification of DNA isolated from the tail at age 2 weeks.

The PCR primers used for genotyping for CTSB[flox/flox] mice were as follows:

5'-ATAAGTGAGCTTTGGAGCGAGTTG-3';
5'-CCACAGTGTCTTCTCTAATCTCCTT-3'

The PCR primers used for genotyping for ggt1cre[+] mice were as follows:

5'-CAGCCTGCTCTAACGGTTTC-3';
5'-CAGGTTCTTGCGAACCTCAT-3'

UOX[+/-] mice were obtained from Cyagen Biosciences (Suzhou, China). Uricase knockout mice (UOX[-/-]) were generated by cross-breeding UOX[+/-] mice. Uricase knockout mice are a model for HUA.

Genotypes were identified by polymerase chain reaction (PCR) amplification of DNA isolated from tail tissue at age 2 weeks.

The PCR primers used for genotyping for UOX[-/-] mice were as follows:
5'-GGCACATGTATCTAACTTTGCCTT-3';
5'-TCTCATGACCTCAACAAATCAATGG-3'

### Mice models
C57BL/6 male mice (20-25 g; aged 8-12 weeks) were used in all experiments. Mice were housed at room temperature (22-25 °C) under standard conditions: humidity of 40-60%, allowing free access to water and food under a 12 h light/dark cycle (light at 8:00 am). The establishment of the mouse model of HUA and the intervention strategy were carried out according to previous reports[64]. The mice were divided into four groups: CTSB[flox/flox] (intraperitoneal injection (i.p.) of 250 mg kg[-1] d[-1] saline for seven consecutive days), CTSB[tecKO] (i.p. injection of 250 mg kg[-1] d[-1] saline for seven consecutive days), CTSB[flox/flox] + HUA (intraperitoneal injection of 250 mg kg[-1] d[-1] PO and drinking 0.1% uric acid water for seven consecutive days), and CTSB[tecKO] + HUA (i.p. injection of 250 mg kg[-1] d[-1] PO and drinking 0.1% uric acid water for seven consecutive days).

To explore the effects of inhibition of CTSB on inflammation and renal uric acid transport, mice were randomly divided into four groups: con (i.p. injection of 250 mg kg$^{-1}$ d$^{-1}$ saline for seven consecutive days), CA074Me (i.p. injection of 10 mg kg$^{-1}$ d$^{-1}$ CA074Me for seven consecutive days), HUA (i.p. injection of 250 mg kg$^{-1}$ d$^{-1}$ PO and drinking 0.1% uric acid water for seven consecutive days), and HUA+CA074Me (i.p. injection of 250 mg kg$^{-1}$ d$^{-1}$ PO and drinking 0.1% uric acid water for seven consecutive days and i.p. injection of 10 mg kg$^{-1}$ d$^{-1}$ CA074Me for seven consecutive days). When inhibiting glycolysis, mice were randomly divided into four groups: con (i.p. injection of 250 mg kg$^{-1}$ d$^{-1}$ saline for seven consecutive days), Lo (i.p. injection of 10 mg kg$^{-1}$ d$^{-1}$ Lo for seven consecutive days), HUA (i.p. injection of 250 mg kg$^{-1}$ d$^{-1}$ PO and drinking 0.1% uric acid water for seven consecutive days), and HUA+Lo (i.p. injection of 250 mg kg$^{-1}$ d$^{-1}$ PO and drinking 0.1% uric acid water for seven consecutive days and i.p. injection of 10 mg kg$^{-1}$ d$^{-1}$ Lo for seven consecutive days).

The urine of mice was collected for 24 hours on the 6th day; the mice were sacrificed after PO or saline injection on the 7th day, and the serum and kidneys were collected.

### Biochemical indicator measurements in mouse serum and urine
The serum samples were sent to the Fudan University Experimental Animal House for measurement of kidney function indicators including Scr, SUA, and blood urea nitrogen (BUN). Urine aliquots were sent to Shanghai Simplegene Medical Laboratory Co., Ltd to determine urinary biochemical indicators such as UUA, Ucr, and urinary microalbumin (U-MALB).

### Soluble uric acid preparation
Uric acid was dissolved in 1 M NaOH at a concentration of 50 mg/ml, as previously described[65]. The solution was examined to ensure that it was free of mycoplasma and filtered (0.22 μm pore size) before use. Crystals were not detectable using polarizing microscopy, nor did they develop during cell incubation.

### Cell culture and treatments
All cell experiments were performed in six-well plates, and cells were cultured in serum-free medium for 24 hours before treatment. Human renal proximal tubular epithelial cells (HK-2) (Cyagen Biosciences, Suzhou, China) were cultured in Dulbecco's modified Eagle's and Ham's F-12 medium (DMEM/F12, Gibco, USA) supplemented with 10% fetal bovine serum (Gibco, USA), 100 U/mL penicillin (Biosharp, China), and 100 mg/mL streptomycin (Biosharp, China) at 37 °C in 5% $CO_2$. When inhibiting CTSB, cells were divided into four groups: con (10 mg/dL NaOH + 0.1% DMSO), CA074Me (10 mg/dL NaOH + 10 μM CA074Me), UA (10 mg/dL uric acid + 0.1% DMSO), and UA+CA074Me (10 mg/dL uric acid + 10 μM CA074Me). When inhibiting glycolysis, cells were divided into four groups: con (10 mg/dL NaOH + 0.1% DMSO), Lo (10 mg/dL NaOH + 100 μM Lo), uric acid (10 mg/dL UA + 0.1% DMSO), and uric acid +CA074Me (10 mg/dL UA + 100 μM Lo).

CTSB-KO HK-2 cells were obtained from Cyagen Biosciences (Suzhou, China) and cultured in Dulbecco's modified Eagle's and Ham's F-12 (DMEM/F12) medium (# 11320074, Gibco, USA) supplemented with 10% fetal bovine serum (# A5669701, Gibco, USA), 100 U/mL penicillin (# BL505A, Biosharp, China), and 100 mg/mL streptomycin (# BL505A, Biosharp, China) in a humidified incubator with 5% CO2. Cells were divided into four groups: WT (WT HK-2:10 mg/dL NaOH), KO (CTSB-KO HK-2:10 mg/dL NaOH), UA (WT HK-2:10 mg/dL uric acid), and KO + UA (CTSB-KO HK-2:10 mg/dL uric acid).

Mouse renal tubular epithelial cells (MRTECs) (Meisen, China) were cultured in Dulbecco's modified Eagle's medium (DMEM)(# 11965092, Gibco; USA) supplemented with 10% fetal bovine serum (# A5669701, Gibco, USA), 100 U/mL penicillin (# BL505A, Biosharp, China), and 100 mg/mL streptomycin (# BL505A, Biosharp, China) in a humidified incubator with 5% CO2. The MRTEC cell treatment methods were the same as the HK-2 cell treatment methods.

### Western immunoblot analysis
Total protein was extracted from kidney and cells in radio-immunoprecipitation assay (RIPA) lysis buffer (#PC101, Epizyme Biomedical Technology, China) with a protease inhibitor cocktail (#GFR101, Epizyme Biomedical Technology, China). A BCA protein assay (#23225, Thermo Fisher Scientific, USA) was used for the quantification of protein concentration. Protein samples were separated by 10% sodium dodecyl sulfate–polyacrylamide gel electrophoresis (#PG2212, Epizyme Biomedical Technology, China) and transferred onto polyvinyl difluoride membranes (#C3317, Millipore, USA). The membranes were incubated overnight at 4 °C with the primary antibodies. Goat anti-rabbit (#7074) or anti-mouse (#7076) IgG antibodies (CST, USA) were used as the secondary antibody at 37 °C for 1 h. Blots were visualized using a Bio-Rad imaging system. ImageJ was used for gray value analysis of the protein bands, and GraphPad Prism 9.0 was used for plotting and statistical analysis.

To ensure accurate normalization, all target proteins and their corresponding loading controls (β-actin) were derived from the same membrane. When probing multiple proteins, membranes were either cut horizontally based on molecular weight and incubated separately with different primary antibodies, or sequentially probed using stripping buffer (#PS107, Epizyme Biomedical Technology, China) to remove previous antibodies before reprobing. In both approaches, care was taken to confirm that the internal controls and target proteins originated from the same blot.

### Quantitative real-time PCR
Total RNA was extracted from the kidney and cultured cells using TRIzol reagent (#15596026, Invitrogen), and cDNA was synthesized using the PrimeScript RT Reagent Kit (#RR037B, Takara). qRT–PCR amplification was performed using ChamQ Universal SYBR qPCR Master MiX (#Q711-02, Vazyme) in the StepOnePlus Real-Time PCR System (Applied Biosystems). All results were calculated using the $2^{-\Delta\Delta CT}$ method. Changes in gene expression were normalized to the levels of β-actin. Primer sequences are listed in Supplementary Table S1.

### RNA-sequencing and data analysis
TRIzol (Invitrogen, Carlsbad, CA, USA) was used to extract total RNA, and libraries were established using the TruSeq Stranded mRNA LTSample Prep Kit (Illumina, San Diego, CA, USA) in accordance with the manufacturer's instructions. The constructed libraries were then sequenced using the Illumina sequencing platform (Illumina HiSeq X Ten or HiSeqTM 2500), which produced paired-end reads of 125 bp/150 bp. Transcriptomic analysis and sequencing was outsourced to OE Biotech Co., Ltd. (Shanghai, China). $P < 0.05$, and fold change (FC) < 0.5 or FC > 2 indicated significant differential expression.

### Metabolomics and data analysis
The mouse kidney samples, 30 mg each, were processed according to the following protocol. Each sample was transferred into a 1.5 mL Eppendorf tube, followed by the addition of a 600 μL methanol-water solution (V:V = 4:1) containing L-2-chlorophenylalanine at a concentration of 4 μg/mL. Two small steel balls were then added, pre-cooled at −40 °C for 2 minutes, to facilitate homogenization. The tubes were subjected to grinding in a mill at 60 Hz for 2 minutes. Subsequently, ultrasonic extraction was performed in an ice-water bath for 10 minutes, followed by a 2-hour incubation at −40 °C to ensure efficient extraction. After centrifugation for 10 minutes at 13000 rpm and 4 °C, 150 μL of supernatant was collected using a syringe. The supernatant was filtered through a 0.22 μm organic phase syringe filter before being transferred to LC vials and stored at −80 °C until LC-MS analysis. For chromatographic analysis, an ACQUITY UPLC HSS T3 column (100 mm × 2.1 mm, 1.8 μm) was utilized with a column temperature maintained at 45 °C. The mobile phase consisted of A- Water (containing 0.1% formic acid) and B- Acetonitrile (containing 0.1% formic acid), with a flow rate of 0.35 mL/min. Injection volumes of 2 μL were used for analysis. Quality control samples (QC) were prepared by mixing equal volumes of the extract from all samples. It is noteworthy that all extraction

reagents were pre-cooled at −20 °C before use to ensure consistency in the experimental conditions.

Data analysis and sequencing were outsourced to OE Biotech Co., Ltd. (Shanghai, China). Variable Importance of Projection (VIP) values derived from the OPLS-DA model served to prioritize the contribution of each variable towards group discrimination. Subsequently, a two-tailed Student's T-test was employed to ascertain the statistical significance of intergroup metabolite variances. Metabolites exhibiting differential expression were identified based on VIP values exceeding 1.0 and $p < 0.05$, thereby ensuring robust selection criteria for further analysis.

### Renal histology

The kidney tissue samples were fixed in 4% paraformaldehyde, dehydrated using an alcohol gradient, and embedded in paraffin. Sections (4 μm) were then cut using a microtome and subjected to hematoxylin and eosin (HE) staining according to the following protocol: Sections were incubated in a 65 °C oven for approximately 1 hour. After removal from the oven, sections were deparaffinized using a staining machine with a xylene-ethanol gradient (from xylene to absolute ethanol). Sections were stained with hematoxylin for approximately 4 minutes, rinsed in tap water for 4 minutes, differentiated in 1% hydrochloric acid alcohol for 3 seconds, rinsed again in tap water for 2 minutes, stained with eosin for 30 seconds, and rinsed once more in tap water for approximately 1 minute. The sections were again dehydrated using an alcohol gradient. Clearing was performed with xylene, and sections were mounted with neutral resin. Tissue staining results were observed under a microscope and photographed.

### Gomori's methenamine silver staining

Kidney tissues were preserved in absolute alcohol, embedded in paraffin wax, and sliced into sections measuring 4–5 μm. The sections were subjected to de-paraffination and subsequent rinsing in absolute alcohol. Subsequently, the sections were subjected to a 60° preheated working methenamine silver solution (#G3030, Solarbio, China) for 30 min, and then incubated in a gold chloride toning agent and an eosin solution in sequence. Observing the slices under a light microscope, it was found that urate salts were stained black.

### Immunofluorescence

After the kidneys were fixed and dehydrated, they were cut into 10 μm sections using a frozen microtome. After blocking with 5% bovine serum albumin, sections were incubated overnight at 4 °C with anti-HK2 (1:100) and PKM2 (1:100) antibodies, followed by incubation with goat anti-rabbit IgG conjugated with Alexa Fluor 488 (#11008) or 594 (#A11012) (1:200, Invitrogen, USA) and LTL (1:200, Vectorlabs, USA) for 1 hour at 37 °C. The nuclei were stained with 4′,6-diamidino-2-phenylindole (DAPI) for 10 minutes at room temperature. The sections were observed under a fluorescence microscope at 40× (Leica, Germany).

HK-2 or MRTEC cells were washed with phosphate-buffered saline (PBS), fixed with 4% polyformaldehyde (PFA) for 30 minutes at room temperature, and then permeabilized with 0.5% Triton X-100 for 10 minutes. After blocking with 5% bovine serum albumin, the cells were kept overnight at 4 °C with anti-CTSB (1:200, #31718 CST, USA) antibody, followed by incubation with goat anti-rabbit IgG conjugated with Alexa Fluor 488 (#11008) or 594 (#A11012) (1:200, Invitrogen, USA) for 1 hour at 37 °C. DAPI was added for 10 minutes at room temperature. The cells were observed under a fluorescence microscope (Leica, Germany).

### Quantification by ELISA

Patient urine was tested using a human CTSB ELISA kit (#D711010, BBI, China) according to the manufacturer's instructions. IL-1β, IL-6 and TNF-α levels were measured using a human IL-1β ELISA kit (#B160351, Hengyuan, China), mouse IL-1β ELISA kit (#HS1069, Hengyuan, China), mouse lL-6 ELISA kit (#KE10007, Proteintech, China) and mouse TNF-α

ELISA kit (#KE10002, Proteintech, China) according to the manufacturer's instructions.

### Assessment of hexokinase activity, pyruvate kinase activity, and lactic acid level

The hexokinase activity test kit (#BC0745), pyruvate kinase activity test kit (#BC0540), and lactic acid level test kit (#BC2235) were purchased from Solarbio (Beijing, China). The tests were conducted according to the manufacturer's instructions.

### Extracellular acidification rate (ECAR) analysis

The glycolysis capacity was obtained through the Seahorse XF® Glycolysis Stress Test Kit [Agilent™, Santa Clara, USA], in line with manufacturer protocol. Briefly, cells were seeded at a density of 2000 cells in a 96-well plate and incubated overnight. The cells were cultivated to 60%–70% confluence, and they were divided into four groups: (1) control group; (2) UA group (3) OE group (4) OE + UA group. The UA group was treated with 10 mg/dl UA for 48 h. The OE group was transfected with lipo3000. And the OE + UA group was cotreated with lipo3000 and UA for 48 h. After washing the cells with Seahorse buffer, 10 mmol/L glucose, 1 mmol/L oligomycin, and 50 mmol/L 2-deoxy-glucose were added to measure the ECAR. ECAR values were calculated and normalized to the cell number. The Agilent Seahorse system we used provides an XF data normalization solution using an automated imaging and cell counting workflow. In our experiments, we set the ECAR value per 10,000 cells for normalization, the Agilent Seahorse XF system automatically calculated and generated the normalized results.

### Statistical analysis

Statistical data are expressed as mean ± standard deviation (SD). Student's $t$ test was used to compare 2 groups. The significance of the differences in mean values between and within multiple groups was examined by one-way ANOVA. $P < 0.05$ was considered statistically significant.

### Data availability

RNA-Seq raw datasets generated from this study were deposited into the SRA database under accession code PRJNA1265931. Numerical source data for graphs are available in Supplementary Data 1. The full metabolite intensity matrix is available in Supplementary Data 2. Uncropped blots are provided in Supplementary Fig. S7. Any additional data used and/or analyzed during the current study are available from the corresponding author upon reasonable request.

### Abbreviations

| | |
|---|---|
| HUA | Hyperuricemia |
| CTSB | Cathepsin B |
| SUA | Serum uric acid |
| Scr | Serum creatinine |
| URAT1 | Urate transporter 1 |
| ABCG2 | ATP-binding cassette subfamily G member 2 |
| GLUT9 | Glucose transporter 9 |
| HK2 | Hexokinase-2 |
| PKM2 | M2-type pyruvate kinase |
| PO | Potassium oxonate |
| FEua | Fractional excretion of uric acid |
| CKD | Chronic kidney disease |
| ESKD | End-stage kidney disease |
| AKI | Acute kidney injury |
| U-MALB | Urinary microalbumin |
| MRTECs | Mouse renal proximal epithelial cells |
| Lo | Lonidamine |
| GLUT1 | Glucose transporter 1 |
| NLRP3 | NOD-like receptor thermal protein domain associated protein 3 |
| ECAR | Extracellular acidification rate |

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

## Acknowledgements

This work was supported by the Excellence Program of Fudan University (JIF163001 to J.X.), Science and Technology Commission of Shanghai Municipality (21Y11904000 to J.X.), Science and Technology Commission of Shanghai Municipality (23141901700 to M.Y.).

## Author contributions

H.L., J.X. and M.Y. conceived and designed the research. H.L., L.N. and D.W. performed experiments. H.L., L.N. and D.W. analyzed data. D.Z., S.T., and R.P. collected patient samples. H.L. and D.W. prepared the figures. H.L. drafted the manuscript. Z.Y., S.Z., M.Y. and J.X. edited and revised the manuscript. J.X. approved the final version of the manuscript.

## Competing interests

The authors declare no competing interests.

## Ethical approval

This study was approved by the Ethics Committee of Huadong Hospital, Fudan University (approval No. 2022K186).

## Informed consent

Written informed consent was obtained from all participants.
