## [Transparent Peer Review file · Communications Biology]

Cathepsin B-dependent glycolysis contributes to reduced renal uric acid excretion in hyperuricemia

Corresponding Author: Dr Jing Xiao

Version 0:

Reviewer comments:

Reviewer #1

(Remarks to the Author)

This study aims to explore the role of cathepsin B (CTSB) in regulating renal uric acid excretion and its impact on hyperuricemia (HUA). The research found that HUA patients have elevated CTSB levels in their urine, and HUA mice show increased serum uric acid (SUA) and renal CTSB expression. CTSB knockout mice demonstrated improved uric acid excretion and lower SUA levels. Further experiments revealed that CTSB deficiency or inhibition reduced glycolysis markers and renal inflammation, enhancing uric acid excretion. These findings highlight the critical role of CTSB in HUA and suggest it as a potential therapeutic target. The following specific comments should be considered by the author for the overall improvement of the manuscript.

1. It is well known that cathepsin B exists in three forms: prepro-cathepsin B, pro-cathepsin B, and mature cathepsin B. In the author's study, are there differences in the expression of these other forms of cathepsin B? If so, why was only the 27 kDa form selected for investigation?

2. The title of the article mentions inflammation, but the manuscript does not address this aspect in the corresponding depth. It could be considered limited to focus on only one inflammatory factor when analyzing the role of cathepsin B in the context of NOD-like receptor (NLR) signaling pathways, especially if transcriptomic data indicate upregulation of the NLR signaling pathway.

3. Fig. 1 k and l, the staining of LTL appears somewhat different.

4. Fig. 2 c, there is no significant difference in SUA levels between CTSB^{flox/flox} mice and CTSB^{teckO} mice. However, the manuscript states that "the average SUA level of CTSB^{flox/flox} mice was higher than that of CTSB^{teckO} mice," which seems to be an inaccurate description of the results.

5. Fig. 2 h, Fig. 6 c, and Fig. 7 f, western blot analyses were performed on four groups of mice, but the grayscale values of the two bands from the same group of mice differ significantly, making it difficult to visually discern expression differences. It is recommended to repeat the experiments and quantify the band grayscale values.

6. Lines 486-512: the author lists the top ten upregulated and downregulated differentially expressed genes in the kidneys of two groups of mice. It is recommended to further explain the relevance of these genes to the study's focus and to integrate the results of the subsequent metabolomics analysis with the transcriptomic data to draw comprehensive conclusions.

7. Fig. 8 a, the black spot on the WB bands affects the grayscale value statistics. Clearer WB result images are needed.

8. In the final part of the results, the author overexpressed CTSB in renal proximal tubular epithelial cells to validate the relationship between CTSB and glycolysis, inflammation, and uric acid. It is recommended to include an experimental group with CA074Me added after CTSB overexpression to provide a more comprehensive explanation.

Minor points:

1. Line 316: The first letter is not capitalized.

2. Lines 335-336 : There are extra blank lines. Please check for consistency in the spacing between each major heading and the following major heading, as well as between paragraphs in the results section. For example, there are blank lines in

lines 400 and 410, but no blank lines between the following paragraphs.

3. Line 385 : The abbreviation "SEM" and its full expression, standard error of the mean, are reversed.

Reviewer #2

(Remarks to the Author)

Reviewer Feedback on "Cathepsin B-dependent glycolysis contributes to renal inflammation and reduced renal uric acid excretion in hyperuricemia"

Dear Editor,

I appreciate the opportunity to review the manuscript titled " Cathepsin B-dependent glycolysis contributes to renal inflammation and reduced renal uric acid excretion in hyperuricemia" by Dr Xiao and colleagues [ID,COMMSBIO-24-4612]. The authors have addressed a compelling topic, and their work could contribute significantly to the field. However, I believe that the manuscript requires substantial revisions before it can be considered for publication. Below, I have detailed my recommendations for both major and minor revisions.

Major Revisions:

1. The authors need to provide a comprehensive characterization of the uricase gene knockout mice phenotypes and baseline parameters. This should include body weight, insulin tolerance test (ITT), glucose tolerance test (GTT), insulin levels, liver function, and kidney function assessments in UOX-KO mice. Additionally, it is crucial to report whether uric acid crystals are observable in the renal tissue.
2. The manuscript currently only presents data on IL-1beta expression. The rationale behind selecting only this inflammatory marker is not clear. The authors should include data on other relevant inflammatory factors such as TNF-alpha and IL-6 to provide a more thorough analysis of the inflammatory profile.
3. For the renal pathology assessment, the inclusion of panoramic scan images is necessary to provide a complete view of the tissue architecture and any potential abnormalities.
4. The authors emphasize the role of CTSB-dependent glycolysis in hyperuricemia yet do not present any experimental data on glycolysis. I recommend they include measurements of the oxygen consumption rate (OCR) and extracellular acidification rate (ECAR) following CTSB knockout or overexpression in a hyperuricemic environment to support their claims. Seahorse XF Pro Analyzer

Minor Revisions:

1. Line 166: The term "co-morbidities" should be corrected to "comorbidities."
2. Lines 171 and 172: The usage of "24-hour" and "24 hours" should be consistent throughout the document.
3. Lines 174 and 184: The temperature notation should be unified to the Celsius symbol (°C).
4. Line 182: The authors should convert rpm to 'g' to account for differences in centrifuge rotor radii and provide a standardized measure.
5. Line 183: There should be a space between the number and the unit (e.g., "50 mL").
6. Lines 220 and 221: The hyphenation in the weight range should be consistent (e.g., "20–25 g").
7. Line 365: The magnification notation "40X" should be corrected to "40x."
8. Throughout the manuscript, the statistical "P" should be consistently italicized.
9. There are inconsistencies between Fig.1j, Fig.2e, and f and their respective textual descriptions that need to be addressed.
10. Quantitative data should be provided for Fig.2h and Fig.7f.
11. The contrast in Fig.2i between CTSBtecKO+HUA and other groups needs to be adjusted for clarity and consistency.
12. In Fig.4b and c, the sample size for the 'con' group is listed as 3, but the statistical analysis reflects a sample size of 2. This discrepancy must be corrected.
13. The formatting of the references should be standardized according to the journal's guidelines.
14. The use of abbreviations in the abstract should be consistent (e.g., "cathepsin B (CTSB)" on lines 41 and 42; "URAT1" on line 46).

The manuscript has the potential to add valuable insights to our understanding of HUA, but these revisions are necessary to ensure the clarity, completeness, and accuracy of the research presented. I look forward to reviewing the revised manuscript.

Sincerely,

Chenggui Zhang

Reviewer #3

(Remarks to the Author)

The manuscript by Lin et al aimed to investigate Cathepsin B-dependent glycolysis contributes to renal inflammation and reduced renal uric acid excretion in hyperuricemia. They identified Cathepsin B (CTSB) as a regulator of reduced renal uric acid excretion in HUA. They showed that HUA patients had higher urinary CTSB levels. HUA mice had increased serum uric acid and renal CTSB. CTSBtecKO HUA mice exhibited reduced renal IL-1β and URAT1, enhancing uric acid excretion and lowering SUA. In vitro and RNA-sequencing analyses revealed that CTSB deficiency or inhibition suppressed glycolysis markers and decreased renal IL-1β and URAT1, increased renal uric acid excretion. They found the relationship

between Cathepsin B and uric acid excretion by regulating inflammation and glycolysis. Their findings are interesting and significant. However, the following items should be clarified before acceptance for publication.

- 1.The manuscript is mainly an observational study, with no in-depth research of mechanisms. They observed several large substance including Cathepsin B, uric acid excretion, inflammation and glucose metabolism (glycolysis) in renal tubular epithelial cell by using the inhibitors and knockout mice. Although many phenotypes and relationships were observed. The molecular mechanisms in detail need further to be clarified. Additional experiments are needed.
- 2.The manuscript only analyze URAT1, but they did not study other transporters including ABCG2, GLUT9, SLC2A9 etc.
- 3.The pathology changes of kidney were caused by uric acid crystals in uricase knockout mice (UOX^{-/-}). How to exclude the effect of uric acid crystals than soluble uric acid on the inflammatory induction of renal tubular epithelium in HUA mice of this study.
- 4.How does Cathepsin B regulate glycolysis to effect on the function of URAT1 to increase the uric acid excretion needs to be further experiment.
- 5.The relationship between Cathepsin B and inflammation in uric acid excretion needs to be further experiment.
6. The weastern blotting strip of URAT1 in Fig7-h is not clear, it is recommended to replace it.

Version 1:

Reviewer comments:

Reviewer #1

(Remarks to the Author)

The authors have addressed all my concerns.

Reviewer #2

(Remarks to the Author)

My concerns have been resolved by the author and it is recommended for publication.

Reviewer #3

(Remarks to the Author)

I have no further comment.

Dear Editor and Reviewers,

We would like to express our sincere gratitude for the insightful and constructive comments on our manuscript entitled “Cathepsin B-dependent glycolysis contributes to reduced renal uric acid excretion in hyperuricemia” (MS ID: COMMSBIO-24-4612). We have carefully considered all the reviewers’ suggestions and have conducted additional experiments, improved data presentation, and substantially revised the manuscript to address the concerns raised.

In the revised version, changes made in response to reviewers’ comments have been clearly highlighted in red in the manuscript. We have also included a detailed point-by-point response below, outlining how each concern has been addressed. We believe these revisions have significantly strengthened the scientific rigor, clarity, and overall impact of our study.

We are truly grateful for the opportunity to revise our manuscript and appreciate your time and efforts in reviewing our work. We hope that the revised version meets the journal’s standards for publication and look forward to your favorable consideration.

Best regards,

Dr. Jing Xiao

Department of Nephrology, Huadong Hospital Affiliated to Fudan University, Shanghai, P.R. China,

Postal Code: 200040

Email: jingxiao13@fudan.edu.cn

Reviewers' comments:

Reviewer #1 (Remarks to the Author):

This study aims to explore the role of cathepsin B (CTSB) in regulating renal uric acid excretion and its impact on hyperuricemia (HUA). The research found that HUA patients have elevated CTSB levels in their urine, and HUA mice show increased serum uric acid (SUA) and renal CTSB expression. CTSB knockout mice demonstrated improved uric acid excretion and lower SUA levels. Further experiments revealed that CTSB deficiency or inhibition reduced glycolysis markers and renal inflammation, enhancing uric acid excretion. These findings highlight the critical role of CTSB

in HUA and suggest it as a potential therapeutic target. The following specific comments should be considered by the author for the overall improvement of the manuscript.

1. It is well known that cathepsin B exists in three forms: prepro-cathepsin B, pro-cathepsin B, and mature cathepsin B. In the author's study, are there differences in the expression of these other forms of cathepsin B? If so, why was only the 27 kDa form selected for investigation?

Author's reply: Thank you for your insightful comment. In our study, we focused on the 27 kDa form of cathepsin B because it represents the mature and enzymatically active form of the protein. This form is directly involved in proteolytic function and therefore more relevant to the physiological or pathological process being studied^{1,2}, making it the most functionally relevant for our investigation. While precursor forms (prepro- and pro-cathepsin B) are important for cathepsin B maturation, their transient nature and potential variability made them less suitable for our specific analysis. Moreover, many previous studies have also focused on the mature form of cathepsin B when investigating its biological functions, further supporting our approach³⁻⁷.

2. The title of the article mentions inflammation, but the manuscript does not address this aspect in the corresponding depth. It could be considered limited to focus on only one inflammatory factor when analyzing the role of cathepsin B in the context of NOD-like receptor (NLR) signaling pathways, especially if transcriptomic data indicate upregulation of the NLR signaling pathway.

Author's reply: We appreciate the reviewer's insightful comments and fully acknowledge that our current analysis of inflammation has certain limitations. Therefore, in the revised manuscript, we have removed "renal inflammation" from the title to more accurately reflect the primary findings of this study and the available data. We plan to further investigate inflammation-related mechanisms in future studies.

3. Fig. 1 k and l, the staining of LTL appears somewhat different.

Author's reply: Thank you for your insightful comment. We acknowledge that the LTL staining in the kidneys of *Uox*^{-/-} mice appears different compared to that in control mice. This difference is attributable to the structural alterations of renal tubules in *Uox*^{-/-} mice. This

morphological change is further corroborated by our hexamine silver staining, which clearly demonstrates abnormal tubular architecture in $Uox^{-/-}$ kidneys. Therefore, we believe that the observed differences in LTL staining reflect genuine pathological remodeling of the tubules rather than technical variation.

4. Fig. 2 c, there is no significant difference in SUA levels between $CTSB^{flx/flx}$ mice and $CTSB^{tecKO}$ mice. However, the manuscript states that "the average SUA level of $CTSB^{flx/flx}$ mice was higher than that of $CTSB^{tecKO}$ mice," which seems to be an inaccurate description of the results.

Author's reply: We sincerely thank the reviewer for pointing out this inaccurate description. Indeed, our original intention was to express that the SUA of $CTSB^{flx/flx}+HUA$ mice was higher than that of $CTSB^{flx/flx}$ mice. We have corrected this statement to avoid any confusion (Results, page 17, Line19).

5. Fig. 2 h, Fig. 6 c, and Fig. 7 f, western blot analyses were performed on four groups of mice, but the grayscale values of the two bands from the same group of mice differ significantly, making it difficult to visually discern expression differences. It is recommended to repeat the experiments and quantify the band grayscale values

Author's reply: We greatly appreciate the reviewer's valuable suggestion. As recommended, we have repeated the Western blot experiments and performed quantitative grayscale

analyses to enhance clarity. The updated Western blot images and corresponding grayscale quantification results are provided in the revised manuscript (Fig. 2j-n, Fig6c-e, Fig.7h-m).

6. Lines 486-512: the author lists the top ten upregulated and downregulated differentially expressed genes in the kidneys of two groups of mice. It is recommended to further explain the relevance of these genes to the study's focus and to integrate the results of the subsequent metabolomics analysis with the transcriptomic data to draw comprehensive conclusions.

Author's reply: We sincerely thank the reviewer for this constructive suggestion. In response, we have now included an additional paragraph in the Discussion section. This integrative analysis strengthens our conclusion that CTSB modulates urate transporter expression and uric acid excretion through glycolysis. The newly added discussion can be found in the revised manuscript (Discussion, Page 25, Line 11-30).

7. Fig. 8 a, the black spot on the WB bands affects the grayscale value statistics. Clearer WB result images are needed.

Author's reply: We sincerely appreciate the reviewer's insightful comment. Following your suggestion, we have repeated the Western blot experiments for Fig. 8a and obtained clearer blot images without any spots. Additionally, we have conducted a new statistical analysis based on the updated grayscale values. The revised figure and quantitative analysis results have now been included in the revised manuscript (Fig.8a).

8. In the final part of the results, the author overexpressed CTSB in renal proximal tubular epithelial cells to validate the relationship between CTSB and glycolysis, inflammation, and uric acid. It is recommended to include an experimental group with CA074Me added after CTSB overexpression to provide a more comprehensive explanation.

Author's reply: We sincerely thank the reviewer for this constructive suggestion. To further validate the relationship between CTSB and glycolysis, inflammation, and uric acid, we have added an additional experimental group in which renal proximal tubular epithelial cells were treated with CA074Me following CTSB overexpression. The results from this experiment have now been included and thoroughly analyzed in the revised manuscript, providing a more comprehensive explanation of the observed effects (Fig.8c-i).

Minor points:

1. Line 316: The first letter is not capitalized.

Author's reply: Thank you very much for pointing out this formatting issue. We have corrected the capitalization in the revised manuscript.

2. Lines 335-336: There are extra blank lines. Please check for consistency in the spacing between each major heading and the following major heading, as well as between paragraphs in the results section. For example, there are blank lines in lines 400 and 410, but no blank lines between the following paragraphs.

Author's reply: We thank the reviewer for highlighting this formatting inconsistency. We have carefully reviewed and adjusted the spacing throughout the Results section, ensuring consistent formatting between major headings and paragraphs. Extra blank lines have been removed, and uniform spacing has been applied across the entire manuscript.

3. Line 385: The abbreviation "SEM" and its full expression, standard error of the mean, are reversed.

Author's reply: We sincerely thank the reviewer for pointing out this mistake. We have corrected the order of the abbreviation and its full expression to "standard error of the mean (SEM)".

Reviewer #2 (Remarks to the Author):

Reviewer Feedback on “Cathepsin B-dependent glycolysis contributes to renal inflammation and reduced renal uric acid excretion in hyperuricemia”

Dear Editor,

I appreciate the opportunity to review the manuscript titled " Cathepsin B-dependent glycolysis contributes to renal inflammation and reduced renal uric acid excretion in hyperuricemia" by Dr Xiao and colleagues [ID,COMMSBIO-24-4612]. The authors have addressed a compelling topic, and their work could contribute significantly to the field. However, I believe that the manuscript requires substantial revisions before it can be considered for publication. Below, I have detailed my recommendations for both major and minor revisions.

Major Revisions:

1. The authors need to provide a comprehensive characterization of the uricase gene knockout mice phenotypes and baseline parameters. This should include body weight, insulin tolerance test (ITT), glucose tolerance test (GTT), insulin levels, liver function, and kidney function assessments in UOX-KO mice. Additionally, it is crucial to report whether uric acid crystals are observable in the renal tissue.

Author's reply: We thank the reviewer for the insightful suggestion. Unfortunately, in our initial experiments, we did not collect baseline metabolic parameters from the UOX^{-/-} mice. To address this issue, we attempted to breed a new cohort of UOX^{-/-} mice. However, due to the high mortality associated with the UOX gene knockout, we were only able to obtain one homozygous KO mouse. This mouse appeared extremely weak and was not suitable for insulin tolerance test (ITT) or glucose tolerance test (GTT) due to ethical concerns.

Nevertheless, we sacrificed the mouse and a WT mouse and collected serum for baseline metabolic assessment. The results are as follows:

	UOX ^{+/+}	UOX ^{-/-}
Body Weight (g)	20.6	20.45
Uric acid (μmol/L)	91.15	661.55
BUN (mmol/L)	7.70	52.33
Scr (μmol/L)	8.92	36.67
ALP (U/L)	446.4	679.3
ALT (U/L)	36.3	33.4
AST (U/L)	107.4	136.8

In addition, to clarify whether the observed renal pathology in UOX^{-/-} mice was caused by uric acid crystals or by soluble uric acid, we regenerated the UOX^{-/-} mouse line and performed detailed histopathological analysis. Specifically, kidney sections were stained with silver hexamine staining, which is commonly used to detect uric acid crystal deposition. No significant uric acid crystals were observed in the kidneys of UOX^{-/-} mice, suggesting that the renal tubular injury in this model is predominantly induced by soluble uric acid.

Furthermore, as a comparison, we also established a previously reported HUA mouse model via co-administration of potassium oxonate and adenine^{8,9}. In this model, silver hexamine staining revealed marked crystal deposition in the renal tubules. These findings collectively support that in our UOX^{-/-} model, the inflammatory response in renal tubular epithelial cells is primarily driven by soluble uric acid, rather than crystal-induced injury.

2. The manuscript currently only presents data on IL-1beta expression. The rationale behind selecting only this inflammatory marker is not clear. The authors should include data on other relevant inflammatory factors such as TNF-alpha and IL-6 to provide a more thorough analysis of the inflammatory profile.

Author's reply: Thank you for your valuable suggestion. In the revised manuscript, we have included measurements of TNF- α and IL-6 to provide a more comprehensive assessment of the inflammatory response, as recommended.

However, due to the initial use of the serum samples for automated biochemical analyses (e.g., serum uric acid, creatinine), the amount of remaining serum stored at -80°C was limited. As a result, we were only able to retrieve sufficient serum from three mice per group for the additional ELISA analyses. The supplementary results are presented below and have been incorporated into the revised manuscript with appropriate explanations (Fig.2g-i, Fig7e-g).

3. For the renal pathology assessment, the inclusion of panoramic scan images is necessary to provide a complete view of the tissue architecture and any potential abnormalities.

Author's reply: We thank the reviewer for this valuable comment. In response, we have included low-magnification images taken using a 5 \times objective lens to provide a more comprehensive view of the renal tissue architecture. Although full digital panoramic scans were not performed, these 5 \times images offer a broad overview of the kidney sections and effectively complement the higher magnification images already presented in the revised manuscript (Fig.2n, Fig.S1h, Fig.S3g).

4. The authors emphasize the role of CTSB-dependent glycolysis in hyperuricemia yet do not present any experimental data on glycolysis. I recommend they include measurements of the oxygen consumption rate (OCR) and extracellular acidification rate (ECAR) following CTSB knockout or overexpression in a hyperuricemic environment to support their claims. Seahorse XF Pro Analyzer

Author's reply: We thank the reviewer for this insightful comment. In our study, we primarily focused on glycolysis. To this end, we have performed extracellular acidification rate (ECAR) measurements using the Seahorse XF Pro Analyzer, which is a well-established method to assess glycolytic function. As shown in Figure 8j (please refer to the revised

manuscript), CTSB OE significantly increased ECAR in the presence of a hyperuricemic stimulus, which is consistent with our findings from the expression levels of key glycolytic enzymes. These data collectively support our conclusion that CTSB promotes glycolysis under hyperuricemic conditions. OCR measurements, which reflect mitochondrial respiration, were not the primary focus of our study but could be investigated in future work.

Minor Revisions:

1. Line 166: The term "co-morbidities" should be corrected to "comorbidities."

Author's reply: We thank the reviewer for identifying this spelling issue. We have corrected "co-morbidities" to "comorbidities" in the revised manuscript.

2. Lines 171 and 172: The usage of "24-hour" and "24 hours" should be consistent throughout the document.

Author's reply: We appreciate the reviewer's careful attention to consistency. We have reviewed and standardized the usage of "24-hour" throughout the manuscript, ensuring consistency.

3. Lines 174 and 184: The temperature notation should be unified to the Celsius symbol (°C).

Author's reply: We thank the reviewer for pointing out this inconsistency. We have unified the temperature notation throughout the manuscript, consistently using the Celsius symbol (°C).

4. Line 182: The authors should convert rpm to 'g' to account for differences in centrifuge rotor radii and provide a standardized measure.

Author's reply: We sincerely appreciate the reviewer's valuable suggestion. We have converted the centrifugation speed from rpm to relative centrifugal force ($\times g$), to account for differences in centrifuge rotor radii and provide a standardized measure.

5. Line 183: There should be a space between the number and the unit (e.g., "50 mL").

Author's reply: We thank the reviewer for noting this formatting issue. We have added the missing space between the number and the unit ("50 mL").

6. Lines 220 and 221: The hyphenation in the weight range should be consistent (e.g., "20–25 g").

Author's reply: We appreciate the reviewer pointing out this inconsistency. We have standardized the hyphenation format in the weight range to "20–25 g".

7. Line 365: The magnification notation "40X" should be corrected to "40 \times ."

Author's reply: We thank the reviewer for pointing this out. The magnification notation has been corrected from "40X" to "40 \times ".

8. Throughout the manuscript, the statistical "P" should be consistently italicized.

Author's reply: We appreciate the reviewer's suggestion. We have carefully revised the manuscript to ensure that the statistical "P" is consistently italicized throughout the manuscript.

9. There are inconsistencies between Fig. 1j, Fig. 2e, and f and their respective textual descriptions that need to be addressed.

Author's reply: Thank you for pointing out the inconsistencies. We have carefully reviewed Fig. 1j, Fig. 2e, and Fig. 2f along with their corresponding textual descriptions.

10. Quantitative data should be provided for Fig. 2h and Fig. 7f.

Author's reply: We appreciate the reviewer's suggestion. Quantitative data have now been provided for Fig. 2h and Fig. 7f, and the corresponding figure legends and main text have been updated accordingly.

11. The contrast in Fig.2i between CTSBtecKO+HUA and other groups needs to be adjusted for clarity and consistency.

Author's reply: Thank you for the helpful comment. We have adjusted the contrast in Fig. 2i to enhance clarity and ensure consistency between the CTSB^{tecKO}+HUA group and other groups.

12. In Fig.4b and c, the sample size for the 'con' group is listed as 3, but the statistical analysis reflects a sample size of 2. This discrepancy must be corrected.

Author's reply: We thank the reviewer for the careful observation. The 'con' group in Fig. 4b and 4c indeed consists of three biological replicates. However, two of the data points are very close in value, which makes them appear overlapped in the graph, possibly giving the impression of only two samples. Here are the raw data of Fig.4b and c:

Figure 4b:

con	HUA
0.40265309	0.59964553
0.43569574	0.61486056
0.43203156	0.55809012

Figure 4c:

con	HUA
0.25134664	0.33352337
0.2524358	0.38188579
0.24970461	0.38188579

13. The formatting of the references should be standardized according to the journal's guidelines.

Author's reply: We appreciate the reviewer's reminder. The references have been reformatted in accordance with the standard *Nature* referencing style, as required by the journal's guidelines.

14. The use of abbreviations in the abstract should be consistent (e.g., "cathepsin B (CTSB)" on lines 41 and 42; "URAT1" on line 46).

Author's reply: Thank you for the helpful suggestion. We have revised the abstract to ensure consistent use of abbreviations. Specifically, full terms followed by abbreviations in parentheses are now used upon first mention (e.g., "cathepsin B (CTSB)" and "urate transporter 1 (URAT1)"), in accordance with standard practice.

The manuscript has the potential to add valuable insights to our understanding of HUA, but these revisions are necessary to ensure the clarity, completeness, and accuracy of the research presented. I look forward to reviewing the revised manuscript.

Author's reply: We sincerely thank the reviewer for the thoughtful and constructive comments. We have carefully addressed all the points raised to improve the clarity, completeness, and accuracy of the manuscript. We believe the revisions have significantly strengthened the quality of our work, and we appreciate the opportunity to resubmit the improved version for your consideration.

Reviewer #3 (Remarks to the Author):

The manuscript by Lin et al aimed to investigate Cathepsin B-dependent glycolysis contributes to renal inflammation and reduced renal uric acid excretion in hyperuricemia. They identified Cathepsin B (CTSB) as a regulator of reduced renal uric acid excretion in HUA. They showed that HUA patients had higher urinary CTSB levels. HUA mice had increased serum uric acid and renal CTSB. CTSB^{teckO} HUA mice exhibited reduced renal IL-1 β and URAT1, enhancing uric acid excretion and lowering SUA. In vitro and RNA-sequencing analyses revealed that CTSB deficiency or inhibition suppressed glycolysis markers and decreased renal IL-1 β and URAT1, increased renal uric acid excretion. They found the relationship between Cathepsin B and uric acid excretion by

regulating inflammation and glycolysis. Their findings are interesting and significant. However, the following items should be clarified before acceptance for publication.

1. The manuscript is mainly an observational study, with no in-depth research of mechanisms. They observed several large substance including Cathepsin B, uric acid excretion, inflammation and glucose metabolism (glycolysis) in renal tubular epithelial cell by using the inhibitors and knockout mice. Although many phenotypes and relationships were observed. The molecular mechanisms in detail need further to be clarified. Additional experiments are needed.

Author's reply: We sincerely appreciate the reviewer's insightful comments regarding the need to elucidate the specific mechanisms underlying the observed phenotype. We fully acknowledge that mechanistic exploration is crucial for advancing the field, and we have devoted substantial efforts over the past seven months to address this key point. To date, we have systematically pursued the following experiments.

Based on our RNA-seq results, we identified *Dpysl4* as a candidate negative regulator of glycolysis that was downregulated following *CTSB* knockout. To explore its functional relevance, we attempted to evaluate *DPYSL4* expression by Western blot using multiple commercially available antibodies. Despite these efforts, we encountered major challenges. Unfortunately, none of the antibodies tested yielded specific bands in kidney or HK-2 cell lysates, and all showed extensive non-specific binding or smearing. As a result, we were unable to validate the differential expression of *DPYSL4* at the protein level.

We respectfully propose that the current findings still provide significant value. In this study, we for the first time propose a potential role of the *CTSB*–glycolysis axis in hyperuricemia-induced renal injury. Although glycolysis has been reported as a mechanistic contributor in various kidney diseases¹⁰⁻¹², to our knowledge, the involvement of *CTSB* in regulating glycolysis under hyperuricemic conditions has not been previously described. In the current manuscript, glycolysis is proposed as a mechanistic link based on both *in vivo* and *in vitro* data. Furthermore, we have discussed other potential mechanisms in the Discussion section, to provide a broader mechanistic context.

In accordance with the reviewer's suggestion, we will continue to explore the precise downstream pathways by which *CTSB* regulates glycolysis, and how glycolysis in turn

modulates urate transporter expression. We appreciate your thoughtful feedback, which has helped us clarify the direction for future mechanistic studies.

2. The manuscript only analyze URAT1, but they did not study other transporters including ABCG2, GLUT9, SLC2A9 etc.

Author's reply: We thank the reviewer for this valuable suggestion. In addition to URAT1, we have now included analyses of other key urate transporters, specifically ABCG2 and GLUT9, in our animal experiments. Western blot data for ABCG2 and GLUT9 have been added to the revised manuscript, and corresponding descriptions have been included in the Results section and figure legends.

3. The pathology changes of kidney were caused by uric acid crystals in uricase knockout mice (UOX^{-/-}). How to exclude the effect of uric acid crystals than soluble uric acid on the inflammatory induction of renal tubular epithelium in HUA mice of this study.

Author's reply: We thank the reviewer for this insightful comment. To clarify whether the observed renal pathology in UOX^{-/-} mice was caused by uric acid crystals or by soluble uric acid, we regenerated the UOX^{-/-} mouse line and performed detailed histopathological analysis. Specifically, kidney sections were stained with silver hexamine staining, which is commonly used to detect uric acid crystal deposition. No significant uric acid crystals were observed in

the kidneys of $UOX^{-/-}$ mice, suggesting that the renal tubular injury in this model is predominantly induced by soluble uric acid.

Furthermore, as a comparison, we also established a previously reported HUA mouse model via co-administration of potassium oxonate and adenine^{8,9}. In this model, silver hexamine staining revealed marked crystal deposition in the renal tubules. These findings collectively support that in our $UOX^{-/-}$ model, the inflammatory response in renal tubular epithelial cells is primarily driven by soluble uric acid, rather than crystal-induced injury.

4. How does Cathepsin B regulate glycolysis to effect on the function of URAT1 to increase the uric acid excretion needs to be further experiment.

Author's reply: We sincerely thank the reviewer for raising this insightful question. In our study, we found that CTSB overexpression enhanced glycolytic activity—evidenced by the upregulation of key enzymes such as HK2 and PKM2—which was accompanied by increased expression of URAT1 and GLUT9 and decreased expression of ABCG2. Conversely, CTSB deficiency suppressed glycolysis and reversed these changes, leading to enhanced renal uric acid excretion.

Nevertheless, as discussed in the revised manuscript, the specific mechanisms by which CTSB-regulated glycolysis influences URAT1 expression remain to be elucidated. We hypothesize that intermediates such as lactate, PFKFB3 activation, and NLRP3 signaling may mediate this process, although further studies are needed to verify these pathways

We have now emphasized this limitation more clearly in the Discussion section and highlighted it as an important direction for future research. We are grateful for the reviewer's thoughtful suggestion, which helps to refine the interpretation and scope of our findings.

5. The relationship between Cathepsin B and inflammation in uric acid excretion needs to be further experiment.

Author's reply: We thank the reviewer for this insightful comment. As noted in the Discussion section, although we observed that CTSB deletion was associated with reduced serum levels of inflammatory cytokines (IL-1 β , IL-6, and TNF- α), the primary focus of our study was to investigate the role of CTSB in uric acid excretion in HUA.

We agree that the potential contribution of inflammation to urate transporter regulation in the context of CTSB activation is of interest and warrants further investigation. However, inflammation-related mechanisms were not the primary focus of this study and were therefore not investigated in depth. We have now emphasized this point more clearly in the revised discussion and consider it an important direction for future studies. We appreciate the reviewer's helpful suggestion, which strengthens the scope and clarity of our work.

6. The western blotting strip of URAT1 in Fig7-h is not clear, it is recommended to replace it.

Author's reply: We thank the reviewer for the helpful suggestion. The Western blot for URAT1 in Fig. 7h has been replaced with a clearer version from a repeat experiment to improve image quality and ensure better visual interpretation. The updated figure has been included in the revised manuscript (Fig.7o).

References:

- 1 Xie, Z. *et al.* Cathepsin B in programmed cell death machinery: mechanisms of execution and regulatory pathways. *Cell Death Dis* **14**, 255,(2023).

- 2 Soudenova, M. *et al.* Behind every smile there's teeth: Cathepsin B's function in health and
disease with a kidney view. *Biochim Biophys Acta Mol Cell Res* **1869**, 119190,(2022).
- 3 Xu, L. B. *et al.* Cathepsin-facilitated invasion of BMI1-high hepatocellular carcinoma cells
drives bile duct tumor thrombi formation. *Nat Commun* **14**, 7033,(2023).
- 4 Lin, Z. *et al.* Cathepsin B S-nitrosylation promotes ADAR1-mediated editing of its own
mRNA transcript via an ADD1/MATR3 regulatory axis. *Cell Res* **33**, 546-561,(2023).
- 5 Song, C. *et al.* Critical role of ROCK1 in AD pathogenesis via controlling lysosomal
biogenesis and acidification. *Transl Neurodegener* **13**, 54,(2024).
- 6 Jeong, J. *et al.* Intracellular calcium links milk stasis to lysosome-dependent cell death
during early mammary gland involution. *Cell Mol Life Sci* **81**, 29,(2024).
- 7 Fang, Q. *et al.* Erbin accelerates TFEB-mediated lysosome biogenesis and autophagy and
alleviates sepsis-induced inflammatory responses and organ injuries. *J Transl Med* **21**,
916,(2023).
- 8 Shi, Y. *et al.* Ubiquitin-specific protease 11 promotes partial epithelial-to-mesenchymal
transition by deubiquitinating the epidermal growth factor receptor during kidney fibrosis.
Kidney Int **103**, 544-564,(2023).
- 9 Hu, Y. *et al.* Autophagy Related 5 Promotes Mitochondrial Fission and Inflammation via
HSP90-HIF-1 α -Mediated Glycolysis in Kidney Fibrosis. *Adv Sci (Weinh)*,
e2414673,(2025).
- 10 Gu, M. *et al.* Protein phosphatase 2A α modulates fatty acid oxidation and glycolysis to
determine tubular cell fate and kidney injury. *Kidney Int* **102**, 321-336,(2022).
- 11 Tiwari, R. *et al.* Postischemic inactivation of HIF prolyl hydroxylases in endothelium
promotes maladaptive kidney repair by inducing glycolysis. *J Clin Invest* **135**,(2024).
- 12 Wei, J. *et al.* Galloflavin mitigates acute kidney injury by suppressing LDHA-dependent
macrophage glycolysis. *Int Immunopharmacol* **150**, 114265,(2025).